# The Geometry of Narrow Fine-Tuning Degradation:
# Trajectory Lock-in and Spectral Bifurcation

Jia Liu [1 2]   Jiaxin Luo [1]   Xinhao Qiu [1]   Yixue Hao [2]   Min Chen [1 3]

## Abstract

Magnitude-based stability proxies such as parameter drift are widely used in narrow-task fine-tuning, yet they do not reliably indicate degradation of broad capabilities. We identify trajectory lock-in: under fixed training conditions for narrow adaptation, the joint evolution of task loss and broad generalization collapses onto a shared low-dimensional degradation curve, so many stabilizers primarily change the rate of progress along this curve rather than altering the curve itself. This yields a drift paradox, in which comparable Euclidean displacement can still correspond to divergent generalization outcomes. To diagnose the underlying structure, we introduce objective-agnostic geometric probes that track the effective update subspace, together with an online harm signal that reflects curvature-dominated channeling toward directions associated with broad degradation. Finally, we show that escaping lock-in requires a spectral bifurcation, namely a qualitative reorientation of the update subspace toward softer curvature modes, thereby improving broad generalization while maintaining matched training performance. We validate these findings across model scales and modalities in narrow-task settings, and report practical deployment procedures and overhead measurements.

## 1. Introduction

In fine-tuning large language models (LLMs), a common stability heuristic treats smaller deviation from the pre-trained weights as safer. Motivated by this view, parameter drift and distance-based regularizers are widely used to constrain

[1]South China University of Technology, Guangzhou, China [2]Huazhong University of Science and Technology, Wuhan, China [3]Pazhou Laboratory, Guangzhou, China. Correspondence to: Min Chen <minchen2012@hust.edu.cn>.

*Proceedings of the 43rd International Conference on Machine Learning*, Seoul, South Korea. PMLR 306, 2026. Copyright 2026 by the author(s).

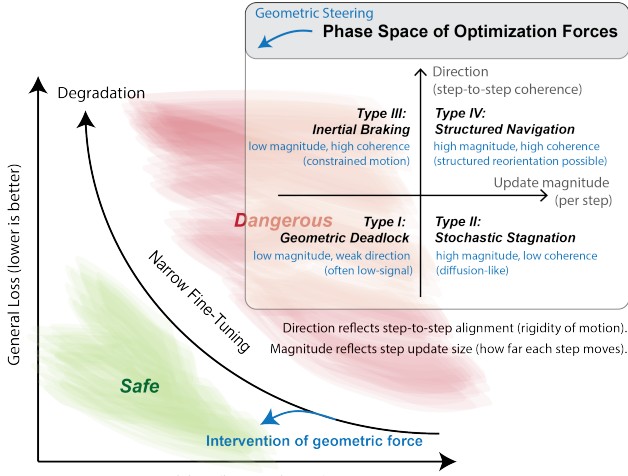

*Figure 1.* **Fine-tuning dynamics as geometric navigation (evaluated settings).** In narrow LoRA fine-tuning, optimization often follows a shared degradation trend in the train–broad plane. The inset summarizes a magnitude–direction phase space that separates four dynamical archetypes.

weight displacement (Li et al., 2025; Xiao et al., 2025; Gururangan et al., 2020).

Empirically, a scalar notion of stability is not predictive: we find runs with comparable Euclidean drift yet different degradation. We term this mismatch the **Drift Paradox**, and use it to motivate diagnostics that capture not only displacement magnitude but also update orientation and effective structure (Figures 1 and 2).

We study LoRA (Zhang et al., 2025b; Paischer et al., 2024; Hu et al., 2022; Zou et al., 2025) and related variants across model scales (1B–8B) and multiple modalities. In the evaluated narrow fine-tuning settings, we repeatedly observe **trajectory lock-in**: over a wide range of optimizers and hyperparameters, accumulated updates concentrate into a low-dimensional geometry and follow a shared degradation trend in the train–broad plane. Along this trend, improvements on the narrow task are coupled with broad capability loss at a roughly stable trade-off slope. Interventions such as null-space projection (Wang et al., 2025a; Fang et al., 2024) and gradient noise injection (Wang et al., 2025c; McCandlish et al., 2018) typically modulate the speed of movement

along the trend, but do not reliably reorient updates away from the degradation direction.

To move beyond drift as a single scalar, we use three objective-agnostic geometric probes: update drift (displacement magnitude), stable update rank (effective update-subspace volume), and directional coherence (trajectory rigidity). These probes make the Drift Paradox measurable and distinguish regimes with similar displacement but different update geometry (Figure 1).

We then test whether the observed lock-in is consistent with spectral bias in the landscape (Ren & Sutherland, 2024; Jacot et al., 2018) by examining the curvature spectrum and the directions emphasized by updates. Our results are consistent with a picture in which curvature modes steer updates toward stiff directions, leading to a low-dimensional attractor. As a probe for directional control, we use an SVD-based structured spectral regularizer that redistributes update energy across modes. In our evaluated settings, it shifts the effective update geometry toward softer directions and reduces broad capability loss at matched training performance relative to baselines; we present it as a diagnostic intervention rather than an optimal method, and do not attribute the effect uniquely to curvature. Here, we use the term *shared degradation path* for the evaluated settings.[1] This work focuses on trajectory-level diagnosis rather than on the general principle that update direction matters. Under matched-loss evaluation, the proposed probes distinguish whether an intervention merely slows movement along a degradation trend or induces a qualitative reorientation of the update trajectory. The SVD-based regularizer is therefore used as a controlled diagnostic probe for studying this distinction, rather than as an optimized production method.

Our contributions are as follows:

- We document cross-scale and cross-modal **trajectory lock-in** in LoRA fine-tuning under evaluated narrow settings, where multiple gradient-based variants follow a shared degradation trend in the train–broad plane.

- We introduce three objective-agnostic geometric probes, namely update drift, stable update rank, and directional coherence, and use them under matched-loss evaluation to quantify the **Drift Paradox** through trajectory-level diagnostics.

- We provide results consistent with a spectral-bias account of lock-in and use a simple SVD-based intervention as a controlled probe to induce trajectory reorientation and mitigate degradation in some regimes, without

claiming optimality.

## 2. Related Work

**Stability via Magnitude Control.** Catastrophic forgetting is commonly addressed by constraining deviation from a pre-trained reference (Kirkpatrick et al., 2017; Zenke et al., 2017; Chaudhry et al., 2018). Methods such as EWC (Kirkpatrick et al., 2017; Huszár, 2018) and norm/penalty regularizers (Xuhong et al., 2018) follow the intuition that limiting $\|\theta - \theta_0\|$ preserves prior capabilities. Similarly, PEFT methods such as LoRA (Hu et al., 2022) and variants (Dettmers et al., 2023; Zhang et al., 2023; Liu et al., 2024) restrict adaptation to low-rank subspaces. A shared limitation is that scalar norms cannot distinguish whether updates concentrate in deleterious versus comparatively benign directions. Our Drift Paradox (Section 3.3) highlights this gap: in the evaluated narrow settings, near-identical drift can coincide with markedly different broad outcomes.

**Geometry of Optimization Trajectories and Landscapes.** Geometric perspectives study curvature, connectivity, and low-dimensional structure of neural optimization (Li et al., 2018b;a; Garipov et al., 2018; Draxler et al., 2018). Most prior work focuses on final minima or connectivity between solutions; in contrast, we study trajectory geometry during adaptation. In the evaluated settings, trajectories often follow a shared degradation trend in the train–broad plane; we use geometric probes to separate interventions that mainly change speed from those that induce qualitative departures.

**Hessian Spectral Structure and Curvature-Dominated Dynamics.** Neural Hessians are often highly anisotropic, with a few large outliers and a near-zero bulk (Sagun et al., 2017; Ghorbani et al., 2019), and such structure has been linked to optimization dynamics and sharpness (Foret et al., 2020; Keskar et al., 2016). We consider a hypothesis that outlier-dominated curvature biases early updates toward stiff directions, contributing to lock-in and low-dimensional concentration. As a lightweight probe, we study an SVD-based spectral regularizer that redistributes update energy across modes; in our settings it can sometimes shift updates toward softer directions and reduce broad loss, without claims of optimality or uniqueness.

**Directional PEFT and Steering Methods.** Recent PEFT studies have explored directionality and spectral structure in LoRA-style adaptation. OPLoRA uses orthogonal projection to reduce interference with pre-trained knowledge (Xiong & Xie, 2026). MiLoRA leverages minor singular components for efficient adaptation (Wang et al., 2025b). C-LoRA studies continual low-rank adaptation under sequential tasks (Zhang et al., 2025a). Our work complements these studies by analyzing trajectory-level dynamics under

---

[1]The *shared degradation path* is an empirical pattern under our protocols on the evaluated LLaMA-family models and tasks; different architectures, objectives, or training regimes may behave differently.

matched-loss evaluation.

# 3. Phenomenological Geometry of Narrow Fine-Tuning Dynamics

We analyze narrow fine-tuning as trajectory evolution in a high-dimensional parameter space. Across the evaluated model scales (1B–8B) and task modalities (Reasoning, Coding, Storytelling), we observe that trajectories often do not freely explore the space, but instead become confined to a shared degradation trend in the train–broad plane. This motivates a phase-space view in which stability is not determined by update size alone, but by the joint evolution of magnitude, direction, and the effective update-subspace volume.

## 3.1. Geometric Probes

To diagnose trajectory structure beyond task loss, we use three complementary geometric probes (implementation details in Appendix F.2). Together they instantiate a magnitude–direction phase space (Figure 2): drift measures update size, coherence measures directional consistency, and stable rank measures subspace volume. Importantly, drift alone is typically insufficient to predict broad-capability outcomes in our evaluated settings; the joint probe patterns support the four dynamical archetypes in Section 3.4.

**Update Drift (Cumulative Magnitude).** We measure the raw scale of adaptation via the cumulative displacement $\|\Delta W_t\|_F$, where $\Delta W_t = W_t - W_0$ denotes the effective weight change relative to initialization (equivalently $\|\mathrm{vec}(\Delta W_t)\|_2$). In LoRA settings, $\Delta W_t$ corresponds to the induced low-rank product $BA$ (up to standard LoRA scaling), rather than the adapter parameters individually; Appendix F.2 specifies the logging construction.

**Stable Rank (Subspace Volume).** To quantify the effective dimensionality of accumulated updates, we monitor the stable rank

$$w_{\mathrm{stable}}(\Delta W_t) \;=\; \frac{\|\Delta W_t\|_F^2}{\|\Delta W_t\|_2^2}, \qquad (1)$$

where $\|\cdot\|_2$ denotes the spectral norm (largest singular value). Larger values indicate update energy distributed across multiple modes, while values near one indicate strong concentration into a single dominant mode. For LoRA, we compute this quantity in a rank-efficient manner without materializing the full weight matrix (Appendix F.2).

**Directional Coherence (Trajectory Rigidity).** We measure step-to-step alignment via the cosine similarity between

consecutive instantaneous updates:

$$c_{\mathrm{step}}(t) = \frac{\langle \delta W_t, \delta W_{t-1}\rangle_F}{\|\delta W_t\|_F \, \|\delta W_{t-1}\|_F}, \qquad (2)$$

where $\delta W_t = W_t - W_{t-1}$ and $\langle A, B\rangle_F = \mathrm{tr}(A^\top B)$. High coherence ($c_{\mathrm{step}} \to 1$) indicates rigid, consistently aligned motion, whereas low coherence indicates rapidly varying directions (diffusion-like motion). When $\|\delta W_t\|_F$ approaches numerical precision, coherence can become unstable; we apply a small-energy guard in logging and report details in Appendix F.2.

## 3.2. Trajectory Lock-in

Figure 3 shows a constraint on adaptation dynamics in the narrow-data regime ($N = 256$). Across Code (MBPP) and Story (TinyStories), Baseline, Null projection, and Gradient Noise follow a similar trend in the train–broad plane: task loss decreases while broad loss increases. Their differences are mainly in how they move along the trend. Baseline progresses faster, Null projection slows the motion, and Noise increases step-to-step variability. However, none of them produces a sustained change in trajectory orientation under the evaluated settings. We refer to this pattern as trajectory lock-in: once optimization enters the dominant degradation trend, lightweight interventions rarely redirect it. In contrast, the SVD-based regularizer can sometimes shift trajectories off the trend, indicating that reorientation is feasible in this regime.

## 3.3. The Drift Paradox

Figures 4 and 5 show the *Drift Paradox*: Baseline and SVD can traverse similar Euclidean distance yet yield sharply different broad-capability outcomes. Distance measures how much the model moved, but it does not encode update orientation. In the evaluated narrow fine-tuning settings, trajectories that follow the shared degradation trend in the train–broad plane incur substantially larger broad-loss increases than runs that achieve partial reorientation, even at comparable drift. The decoupling persists across $N \in [64, 4096]$, so stability is better reflected by geometry-aware signals of orientation and concentration than by drift alone (Appendix K).

## 3.4. Dynamical Archetypes

To summarize optimization behaviors in a way consistent with the phase-space inset of Figure 1, we classify trajectories by two observable quantities: update magnitude per step and step-to-step directional coherence. Concretely, the horizontal axis corresponds to the typical per-step update size (e.g., $\|\delta W_t\|$), and the vertical axis corresponds to the coherence statistic defined in Eq. (2). We use stable-rank patterns (Section 3.1 and Appendix J) as supporting evi-

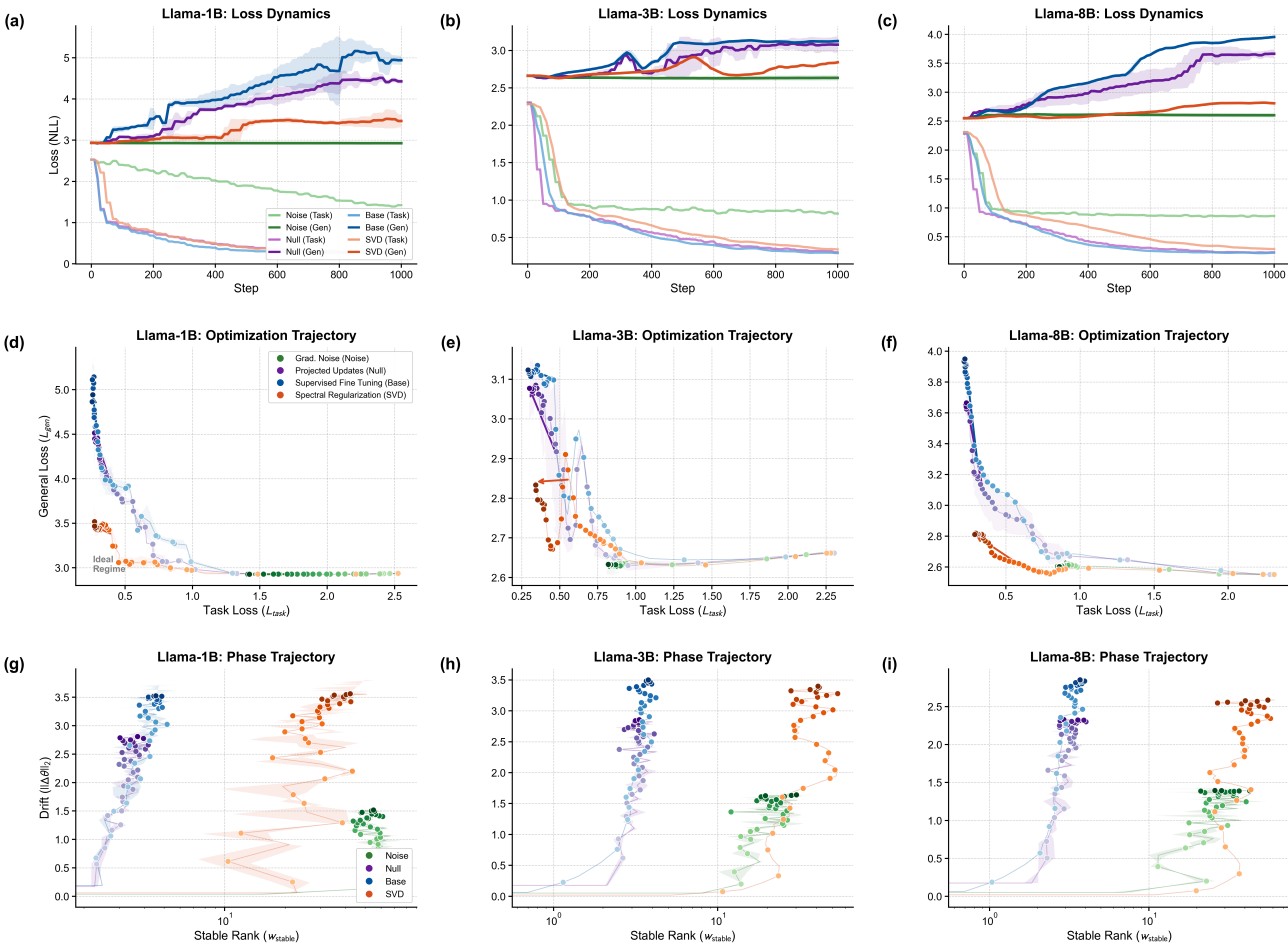

*Figure 2.* **Geometric phase structure of LoRA fine-tuning (evaluated settings). (Top Row)** Loss dynamics across LLaMA-1B/3B/8B. Solid lines show broad/general loss ($L_{\text{gen}}$; evaluation-only), and faded lines show task loss ($L_{\text{task}}$). **(Middle Row)** Trajectories in the loss plane ($L_{\text{gen}}$ vs. $L_{\text{task}}$). Baseline (Blue) and Null projection (Purple) typically progress along a shared degradation trend; an SVD-based spectral regularizer (Orange) is used as a lightweight probe that can sometimes induce a qualitative departure at comparable training performance. **(Bottom Row)** Trajectories in a geometric phase plane (stable-rank volume $w_{\text{stable}}$ vs. cumulative drift). The SVD-based probe often maintains broader update-subspace volume than baseline-like runs, whereas gradient noise can yield diffusion-like motion with weak directional coherence (see Section 3.4).

dence to interpret whether updates concentrate into a narrow subspace.

**Type I: Geometric Deadlock.** Low update magnitude with weak direction indicates a low-signal regime where progress is limited and updates lack a consistent orientation. In the evaluated settings, this pattern often co-occurs with strong concentration in the effective update (low stable rank), and is associated with unfavorable broad outcomes once the trajectory aligns with the degradation trend.

**Type II: Stochastic Stagnation.** High update magnitude with low coherence corresponds to diffusion-like motion. The trajectory moves substantially, but directions vary rapidly across steps, so it does not form a stable reorientation relative to the dominant trend.

**Type III: Inertial Braking.** Low update magnitude with high coherence reflects constrained motion. Updates remain aligned across steps, but their scale is small, so the effect is primarily to slow progression rather than to change orientation.

**Type IV: Structured Navigation.** High update magnitude with high coherence indicates sustained, structured motion that can support reorientation of the trajectory. In some evaluated settings, an SVD-based spectral regularizer serves as a lightweight probe that can instantiate this regime, without implying optimality or universality.

This taxonomy describes what regimes exist; Section 4 explains why they arise.

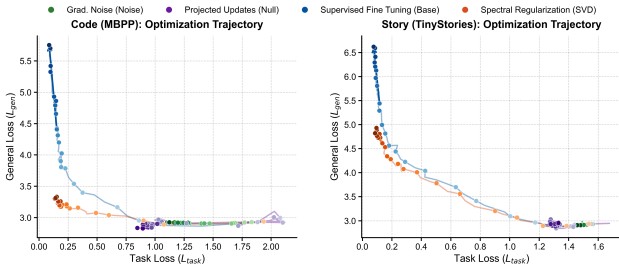

*Figure 3.* **Trajectory consistency across modalities (N=256; evaluated settings).** Train–broad trajectories for Code (MBPP) and Story (TinyStories). Despite distinct data distributions, Baseline (Blue), Null projection (Purple), and Gradient Noise (Green) often evolve along a similar degradation trend in the loss plane. An SVD-based spectral regularizer (Orange), used here as a lightweight probe, can sometimes induce a qualitative departure toward lower broad loss at comparable task loss.

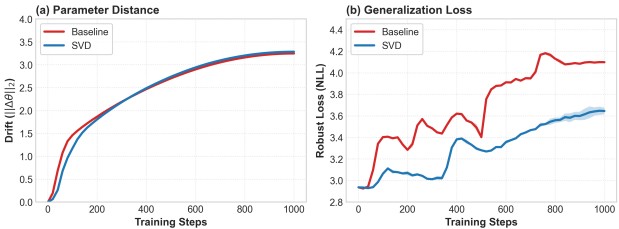

*Figure 4.* **The Micro-Illusion (Temporal).** Baseline (Red) and SVD (Blue) travel nearly identical Euclidean distances ($\|\Delta W\|_2 \approx 3.2$) on LLaMA-1B. However, Baseline loss explodes while SVD loss remains low, confirming that scalar distance is a misleading indicator of safety.

# 4. Mechanistic Interpretation: The Geometry of Degradation

Section 3 showed that fine-tuning trajectories are consistently confined to a shared degradation path unless a structured geometric bifurcation occurs. Here we provide a mechanistic interpretation of this behavior by analyzing the local spectral geometry of gradient updates around the pre-trained initialization. Throughout, we treat the curvature account as an organizing hypothesis that yields testable signatures, rather than a unique causal explanation of all forgetting phenomena.

We use the initialization Hessian as a local probe for early-time update geometry in the effective trainable space (e.g., $\Delta W_t$ for LoRA). We do not claim the quadratic approximation at $\theta_0$ explains the full fine-tuning trajectory, especially after leaving the local neighborhood or entering weak-gradient/saturation regimes. We therefore assess the curvature account via falsifiable signatures in our evaluated settings; if they fail, alternative mechanisms such as data homogeneity or activation/gradient saturation may instead more directly explain the observed collapse.

Our central hypothesis is that the shared path is consistent

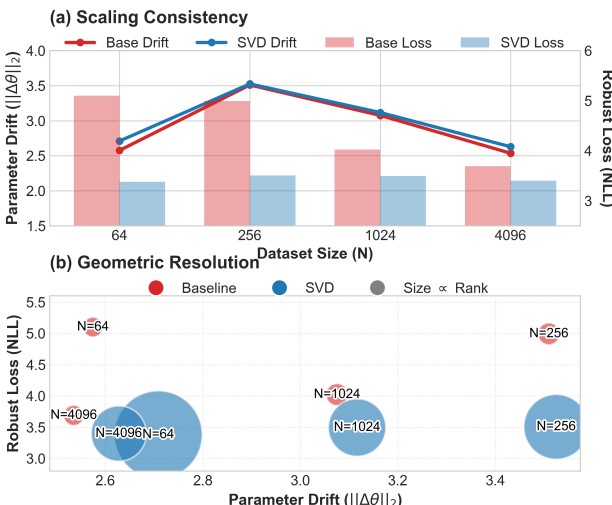

*Figure 5.* **The Macro-Decoupling (Scaling). (a)** Across data regimes ($N \in [64, 4096]$), the "Scissors Gap" persists: equal drift yields divergent outcomes. **(b)** Geometric Resolution: Robustness correlates with subspace volume (Rank, bubble size) rather than displacement magnitude.

with anisotropic local curvature at initialization: a small set of stiff directions can dominate early optimization and bias updates toward a low-dimensional subspace. In parameter-efficient settings, this hypothesis should be interpreted in the effective update space induced by the trainable subspace (e.g., $\Delta W_t$ in LoRA), rather than the full parameter space in isolation. This hypothesis yields two concrete, testable predictions (P1–P2), formalized in Section 4.4.

### 4.1. Physical Intuition: The Canyon and the Ridge

We interpret the landscape geometry physically: the parameter space is a terrain where Hessian eigenvalues correspond to slope steepness.

**The Canyon (Lock-in):** The stiff subspace $\mathcal{V}_{stiff}$ acts as a steep-walled canyon. Standard optimization acts like a ball channeled by gravity into this ravine. Once inside, the trajectory is locked to the valley floor; it can minimize task loss (move forward) but cannot escape the shared degradation path. This mirrors the *Trajectory Lock-in* observed in Section 3.2.

**The Ridge (Bifurcation):** SVD regularization applies a lateral force against this pull. By penalizing energy concentration, it steers the trajectory out of the canyon and onto "gentle ridges" (soft modes). Here, the model learns the task while maintaining distance from the stiff directions associated with forgetting. This distinction resolves the *Drift Paradox*: equal drift distance can lead to divergent outcomes depending on whether one navigates the canyon floor or the open ridge.

## 4.2. Hessian Anisotropy and Stiff Subspaces

Consider the local quadratic approximation of the task loss around the pre-trained parameters $\theta_0$:

$$\mathcal{L}_{\text{task}}(\theta) \approx \mathcal{L}_{\text{task}}(\theta_0) + \nabla \mathcal{L}_{\text{task}}(\theta_0)^\top (\theta - \theta_0) \\ + \frac{1}{2}(\theta - \theta_0)^\top H_{\text{task}}(\theta - \theta_0), \quad (3)$$

where $H_{\text{task}} = \nabla^2 \mathcal{L}_{\text{task}}(\theta_0)$. Empirically, pre-trained models often exhibit anisotropic curvature: the spectrum of $H$ contains a small number of dominant eigenvalues, while many remaining directions are comparatively flat. As a result, gradient-based updates near $\theta_0$ can acquire large components along the corresponding eigenvectors, especially under step sizes that amplify stiff-mode responses. We refer to the span of these dominant modes as a stiff subspace. In PEFT regimes, updates are restricted to a trainable subspace, so we consider the effective stiff directions after restricting or projecting curvature to the induced update space; Appendix L details the construction and sensitivity to eigenvalue cutoffs and optimizer variants. This local analysis does not claim a global characterization of the landscape, but motivates a mechanism for early-time channeling.

## 4.3. Low-Rank Attractor

Dominance of a stiff subspace can induce a low-rank attractor in the effective update geometry. As updates accumulate along a small number of directions, the effective dimensionality of the cumulative update $\Delta W_t$ (e.g., the induced LoRA update in weight space) can concentrate. This is reflected empirically by decreasing stable rank ($w_{\text{stable}}$) and increasing directional coherence ($c_{\text{step}}$) in the lock-in regime.

Crucially, while we allocate a LoRA rank of $r = 128$, the observed effective rank typically remains below 32 throughout training. This substantial gap confirms that the low-dimensional confinement is driven by the intrinsic geometry of the loss landscape (the attractor), rather than being an artifact of the LoRA rank constraint.

Once trajectories enter this attractor, further optimization primarily redistributes energy within the same subspace, producing the observed shared degradation path in the train–broad plane. Because the dominant modes are anchored at initialization, trajectories from heterogeneous tasks can exhibit similar geometric structure despite differing objectives. At the same time, low-dimensional behavior can also arise from qualitatively different regimes, including weak-gradient or saturation effects that shrink the effective Jacobian; our goal is to distinguish these possibilities using the signatures in Section 4.4 rather than attributing all cases to curvature alone. Connections between this attractor view and the dynamical archetypes in Section 3.4, as well as the geometric role of different intervention regimes, are discussed in Appendix M.

## 4.4. Empirical Signatures: Two Predictions

The attractor hypothesis yields two falsifiable predictions, each associated with a concrete diagnostic quantity. These predictions serve as the empirical contract between the proposed mechanism and the experiments that follow.

**P1: Static Alignment.** If fine-tuning trajectories are constrained by initialization curvature, cumulative updates should exhibit elevated alignment with a stiff subspace identified at $\theta_0$ (within the corresponding effective update space used for logging).

Let $\Delta w_t = \text{vec}(\Delta W_t) \in \mathbb{R}^d$ denote the flattened cumulative update, and let $\mathcal{V}_{\text{stiff}} = [v_1, \ldots, v_k] \in \mathbb{R}^{d \times k}$ contain the top-$k$ eigenvectors of the (possibly restricted) Hessian operator used in our probe. We measure alignment by

$$\alpha_t = \frac{\sum_{i=1}^k (v_i^\top \Delta w_t)^2}{\|\Delta w_t\|_2^2} = \frac{\|\mathcal{V}_{\text{stiff}}^\top \Delta w_t\|_2^2}{\|\Delta w_t\|_2^2}. \quad (4)$$

**Prediction.** Baseline fine-tuning tends to maintain higher $\alpha_t$, while runs exhibiting a qualitative departure in the train–broad plane show reduced alignment with the stiff subspace.

**P2: Dynamic Channeling.** If updates are channeled through a stiff subspace, task gradients should systematically interfere with gradients associated with broad capabilities.

We quantify this interference online by measuring cosine alignment between the task gradient $g_{\text{task}}^{(t)}$ and a fixed broad-probe gradient $g_{\text{probe}}$ defined under the evaluation protocol (Appendix F.2 specifies how $g_{\text{probe}}$ is computed and held fixed):

$$h_t = \frac{g_{\text{task}}^{(t)\top} g_{\text{probe}}}{\|g_{\text{task}}^{(t)}\|_2 \|g_{\text{probe}}\|_2}, \quad \text{Abs-Harm}_t = |h_t|. \quad (5)$$

**Prediction.** Standard fine-tuning exhibits increasing cumulative Abs-Harm, coinciding with broad degradation; geometric control suppresses this accumulation in regimes where directional reorientation occurs.

A geometric resolution of the Drift Paradox and its relation to subspace orientation is deferred to Appendix K.

## 5. Experimental Verification

We test the stiff-subspace channeling hypothesis (Section 4) with a concrete question: in the evaluated narrow LoRA settings, can an intervention change the direction of the fine-tuning trajectory so that it departs from the observed degradation trend, without confounding improvements with slower optimization? We adopt a Narrow–Broad protocol:

optimization is performed on a narrow task distribution, while general capability is monitored on a disjoint broad chat set that never contributes gradients. We compare four regimes: Baseline LoRA fine-tuning, Gradient Noise (unstructured variance injection), Null-space Projection (removing dominant update components as a geometric brake), and an SVD-based spectral regularizer. Throughout, we enforce matched-train-loss evaluation: methods are compared at the same narrow-task loss $\ell^*$ (via interpolation along logged trajectories), and we report $\mathcal{L}_{\text{general}}(\ell^*)$. Beyond loss curves, we log drift $\|\Delta W_t\|_F$, stable-rank-like volume $w_{\text{stable}}(\Delta W_t)$, and a step-to-step direction statistic (logged as $g_{\text{cos}}$ as a practical proxy; details in Appendix C). Full implementation details are provided in Appendix C.

### E1: Controller Sensitivity   (SVD strength calibration)

**Question.** In the evaluated sweep, does varying the regularization strength $s$ produce a predictable, strength-dependent departure under matched-loss comparisons? **Evidence.** Figure 6 shows a graded transition: weak control remains close to the baseline trend; moderate control attains lower $\mathcal{L}_{\text{general}}$ at comparable task loss; very strong control yields the largest deviation but also slows task-side progress. **Implication.** This pattern is consistent with strength-dependent reorientation of update geometry within this simple regularizer family, with an operating window where broad loss improves without a large loss in task progress. To verify that these gains are not sensitive to specific hyperparameter choices, we provide a parameter sweep in Appendix P. Results in Table 12 show that the geometric steering effect is maintained across a wide range of regularization strengths ($s \in [0.5, 1.5]$), distinguishing our method from the instability often observed in standard baselines.

### E2: Temporal Hysteresis   (finite rescue window)

**Question.** In the evaluated setting, does the effectiveness of the same rescue depend on optimization history? **Evidence.** In Figure 7, early rescue changes the trajectory direction and suppresses broad degradation, whereas late rescue produces little deviation from the baseline trend despite continued task learning; the protection rate $\rho(t_{\text{rescue}})$ decreases as rescue is delayed. **Implication.** These results are consistent with a history-dependent effect: beyond a stage, local control produces limited directional change.

### E3: Mechanistic Probing   (Abs-Harm as an online channeling signature)

**Question.** In the evaluated runs, does Abs-Harm behave as an online signature of persistent routing of task updates into directions that correlate with broad degradation? **Evidence.** Figure 8 shows that baseline training accumulates high cumulative Abs-Harm that tracks broad loss over time,

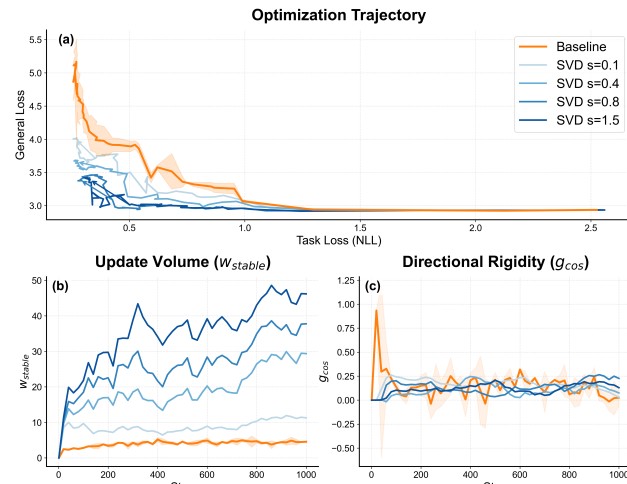

*Figure 6.* **Controller sensitivity to SVD strength $s$ (main evidence). (a)** Train–broad trajectories: weak $s$ stays near the baseline trend; moderate $s$ produces a clear departure with lower general loss at matched task loss; very strong $s$ shows reduced task-side progress. **(b)** Stable-update volume $w_{\text{stable}}$ increases with $s$ in this sweep, indicating that update energy is less concentrated in a single mode. **(c)** The directional statistic $g_{\text{cos}}$ is more stable under SVD regularization. Shaded regions denote standard deviation across seeds.

| Metric | Baseline | SVD |
|---|---|---|
| Mean Abs-Harm ($\downarrow$) | $0.020 \pm 0.003$ | $0.012 \pm 0.001$ |
| AUC($|h|$) ($\downarrow$) | $20.19 \pm 2.82$ | $12.34 \pm 0.60$ |
| Final $\mathcal{L}_{\text{general}}$ ($\downarrow$) | $4.54 \pm 0.34$ | $3.42 \pm 0.03$ |

*Table 1.* **Cross-seed summary of Abs-Harm statistics.** SVD reduces both the instantaneous magnitude (Mean) and the long-horizon accumulation (AUC) of this channeling indicator in the evaluated sweep. Cumulative Abs-Harm is predictive of broad loss in Figure 8.

while the SVD regularizer suppresses this accumulation; Table 1 summarizes the cross-seed reduction in both mean and AUC. **Implication.** Abs-Harm provides an online diagnostic tied to update direction, rather than end-of-run performance alone.

### E4: Alternative Exclusion   (suppression vs. steering)

**Question.** In the evaluated narrow setting, can the gains from SVD be explained by "more stability" alone (e.g., smaller drift or higher $w_{\text{stable}}$), or do they require directional change? **Evidence.** Table 2 contrasts Gradient Noise and SVD: noise preserves broad capability mainly by suppressing task learning (underfitting), whereas SVD retains near-baseline task progress while reducing broad loss under matched-loss evaluation. **Implication.** This comparison is consistent with a directional account: improvements are associated with reorientation rather than magnitude reduction alone.

| Regime | Task Loss $\mathcal{L}_{\text{task}}(t_f)\downarrow$ | Broad Loss $\mathcal{L}_{\text{general}}(t_f)\downarrow$ | Mean Abs-Harm $\downarrow$ | Stability Mode | Trajectory Effect |
|---|---|---|---|---|---|
| **Baseline** | **0.28** | 5.10 | 0.017 | Unconstrained | Lock-in to shared path |
| **Gradient Noise** | 0.97 | **2.99** | **0.007** | Unstructured suppression | Stabilized without bifurcation |
| **SVD regularizer** | **0.31** | 3.51 | 0.012 | Structured steering | Bifurcation from shared path |

*Table 2.* **Disambiguating stability mechanisms via the plasticity–stability trade-off.** In this sweep, Gradient Noise retains broad capability primarily by suppressing task learning through unstructured variance inflation, whereas the SVD regularizer maintains near-baseline task progress while reducing harmful alignment indicators, consistent with directional reorientation.

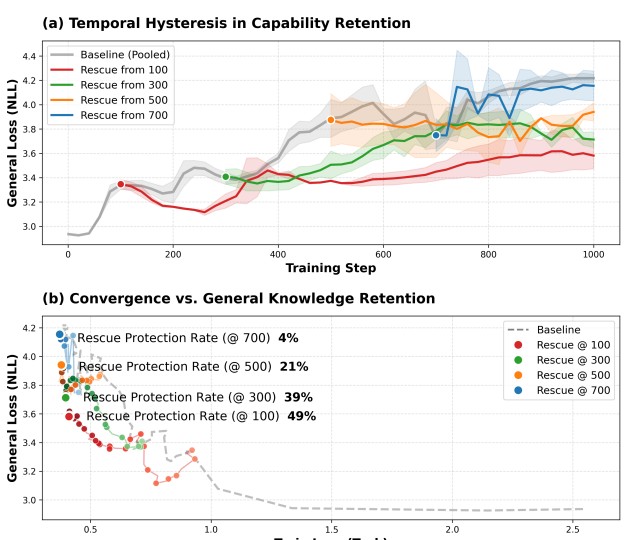

*Figure 7.* **Temporal hysteresis in rescue dynamics. (a)** General loss over training. Earlier rescue suppresses degradation, while late rescue yields limited deviation from the baseline run. **(b)** Optimization trajectories in the train–general loss plane. Initiating rescue at different points along the same baseline trajectory yields different outcomes, indicating a finite window where reorientation is more effective. Protection rates are annotated for each rescue timing.

**Deployment Feasibility** The SVD regularizer adds modest overhead (about $6.6\%$ time/step with no additional peak VRAM); detailed benchmarking is reported in Appendix D.

# 6. Discussion

This work reframes fine-tuning degradation as a trajectory geometry phenomenon. In narrow settings, diverse runs concentrate into a shared degradation trend, coupling task improvement with capability loss even at comparable Euclidean drift (the *Drift Paradox*). This motivates monitoring update orientation, not just magnitude.

We distinguish stability by *suppression* from stability by *steering*. Unstructured interventions (e.g., gradient noise) can preserve broad capabilities by slowing learning, often sacrificing task plasticity. By contrast, structured regulariza-

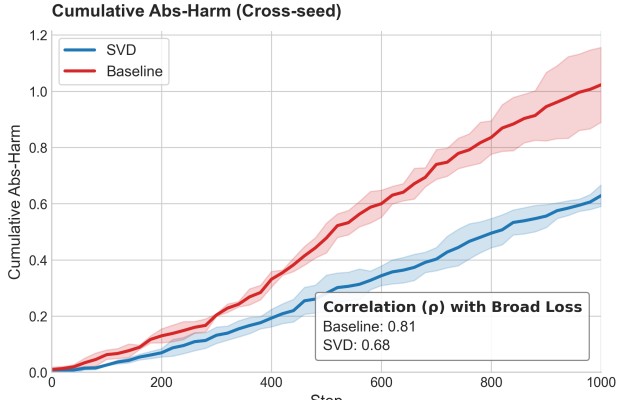

*Figure 8.* **Abs-Harm as an online channeling probe.** The plot shows cumulative Abs-Harm ($\sum_t |h_t|$) averaged across seeds. Baseline (red) exhibits sustained accumulation that correlates ($\rho = 0.81$) with broad loss in this sweep. The SVD regularizer (blue) suppresses long-horizon accumulation. Shaded regions denote $\pm 1$ standard deviation.

*Table 3.* **Overhead Analysis.** Comparison of relative training latency and peak memory usage.

| Method | Time/Step | Rel. Overhead | Peak Mem |
|---|---|---|---|
| Baseline (LoRA) | 328.7 ms | – | 4250 MB |
| Gradient Noise | 332.0 ms | +1.0% | 4250 MB |
| Null-Space Proj. | 382.1 ms | +16.2% | 4252 MB |
| SVD Control | 350.4 ms | +6.6% | 4250 MB |

tion can reorient update geometry, reducing degradation at matched training performance. These differences necessitate probes that track update magnitude, subspace volume, and directional coherence. Mechanistically, we discuss a spectral-bias hypothesis where early updates are channeled into stiff directions, causing lock-in, though data homogeneity may also produce low-dimensional dynamics. Accordingly, "low-rank" behavior can reflect distinct regimes separable only by observables beyond scalar drift.

The present evidence should be interpreted within a bounded scope. Our main experiments focus on LLaMA-family models under LoRA-based narrow-task adaptation, with additional stress tests used to clarify operating boundaries. The

results should not be read as a universal claim over all PEFT families, full-parameter fine-tuning regimes, or mixture-of-experts architectures. For example, in micro-dataset full fine-tuning ($N = 64$, Appendix G), spectral regularization fails, suggesting that task-fitting and broad-degradation directions can become effectively collinear and overwhelm first-order geometric steering. These observations suggest that the proposed probes are best viewed as diagnostics for identifying the current adaptation regime, rather than as guarantees of universal steering effectiveness. Additional small-scale scope checks, including AdamW optimization, learning-rate variation, selective tuning, and Hessian-proxy overlap, are summarized in Appendix H.

Finally, while our analysis is method-agnostic, probe definitions matter for reproducibility. Beyond post-hoc analysis, the online Abs-Harm signal may also serve as a practical warning indicator for early stopping or rescue triggering when broad-degradation channels begin to accumulate. Controlled studies varying optimizer, learning rate, and data diversity remain useful for further disambiguating steering from suppression.

## 7. Conclusion

We presented a geometric analysis of fine-tuning degradation, identifying a shared trajectory lock-in driven by alignment with stiff loss directions. By introducing probes like effective gradient rank, we made this hidden geometry observable during training. Crucially, our experiments show that preventing degradation is not solely about stability: while unstructured variance inflation merely suppresses learning, structured spectral steering enables a bifurcation away from the shared degradation path, achieving a favorable plasticity–stability trade-off. These findings suggest that future methods should prioritize directional control over scalar constraints, highlighting optimization geometry as a first-class object for understanding and controlling adaptation in large language models.

## Acknowledgements

This work was supported in part by the National Natural Science Foundation of China under Grant No. 52539005, in part by the National Key Research and Development Program of China under Grant No. 2025YFE0213400, in part by the National Natural Science Foundation of China under Grant No. 62276109, and in part by the Interdisciplinary Research Program of Huazhong University of Science and Technology under Grant No. 5003210069.

AI assistants were used in a limited manner for language polishing and editorial refinement of the manuscript, but not for generating results, conducting analyses, or making scientific decisions.

## Impact Statement

This paper studies optimization dynamics in fine-tuning large language models, with the goal of improving the reliability and controllability of narrow-task adaptation. By characterizing trajectory lock-in and conditions for spectral reorientation, our findings may help reduce unintended degradation of broad generalization and other safety-relevant behaviors during adaptation. As with many advances in model adaptation, the techniques and insights could be used in dual-use settings; however, our intent and evaluation focus on robustness and predictability rather than capability escalation. We do not claim to anticipate all downstream societal impacts, but we expect the primary effect to be improved monitoring and mitigation of harmful fine-tuning dynamics.

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

# A. Nomenclature and Notation

Table 4 summarizes the notation used for optimization dynamics, geometric probes, and spectral analysis.

*Table 4.* **Table of Notations.** Symbols used for optimization dynamics, geometric probes, and spectral analysis.

| Symbol | Description | Reference |
|---|---|---|
| *Model and Optimization Context* | | |
| $\theta, \theta_0$ | Model parameters (current / initialization). | Sec. 4.2 |
| $t, \eta$ | Training step index; Learning rate. | Sec. 3.1 |
| $N$ | Narrow-task dataset size. | Fig. 5 |
| $W_t, \Delta W_t$ | Trainable matrix state / Cumulative change. | Sec. 3.1 |
| $\mathcal{L}_{\text{task}}, \mathcal{L}_{\text{gen}}$ | Training loss / General capability monitoring loss. | Sec. 3.3 |
| $\ell^*$ | Target task-loss level for matched-loss comparison. | App. C.4 |
| $g_{\text{task}}^{(t)}, g_{\text{probe}}$ | Task gradient / Reference broad-probe gradient. | Sec. 4.4 |
| *Geometric Probes* | | |
| $\|\Delta W_t\|_2$ | Update drift norm used for displacement tracking. | Sec. 3.1 |
| $w_{\text{stable}}$ | Stable update rank proxy for subspace volume. | Sec. 3.1 |
| $c_{\text{step}}, g_{\text{cos}}$ | Directional coherence (theoretical / logged). | Sec. 3.1 |
| $h_t$, Abs-Harm | Signed / Magnitude of gradient alignment signal. | Sec. 4.4 |
| $\text{AUC}(|h|)$ | Cumulative channeling summary over training. | Sec. 4.4 |
| *Mechanistic and Spectral Analysis* | | |
| $H_{\text{task}}, H_{\text{gen}}$ | Hessian of task / broad loss at initialization. | Sec. 4.2 |
| $\{v_i\}, k$ | Leading eigenvectors and truncation level. | Sec. 4.4 |
| $\mathcal{V}_{\text{stiff}}$ | Stiff subspace induced by dominant curvature. | Sec. 4.4 |
| $\mathcal{M}, \alpha_t$ | Low-rank attractor / Alignment score with $\mathcal{V}_{\text{stiff}}$. | Sec. 4.3 |
| *Interventions and LoRA Configuration* | | |
| $s$ | SVD control strength. | App. C.4 |
| $t_{\text{rescue}}, \rho$ | Intervention step / Rescue protection rate. | App. C.5 |
| $r, \alpha$ | LoRA rank and scaling hyperparameter. | App. F.1 |
| $A, B$ | Low-rank adapter matrices. | App. F.1 |

# B. Glossary of Core Concepts

Table 5 synthesizes the conceptual framework used to describe the geometric dynamics of narrow fine-tuning. It maps distinct phenomenological observations to their underlying mechanisms and intervention regimes.

*Table 5.* **Glossary of Core Concepts.** Unified terminology for geometric phenomena, mechanisms, and intervention regimes.

| Unified Concept | Synonyms / Related Terms | Definition & Context |
|---|---|---|
| *Phenomenology: Observable Dynamics* | | |
| **Shared Degradation Path** | Shared Manifold; Shared Curve | An empirically consistent trajectory manifold in the task–broad loss plane. Under narrow fine-tuning, progress along this path enforces a strict trade-off between task improvement and general capability loss. *Note: "shared" refers to cross-setting consistency within the evaluated architectures and optimizers.* |
| **Trajectory Lock-in** | Manifold Confinement; Channeling; Geometric Deadlock | A dynamical state where optimization becomes confined to the degradation path, precluding the discovery of orthogonal, capability-preserving directions. Diagnosed by high directional coherence ($c_{\text{step}} \to 1$) and collapsing effective update rank. |
| **Drift Paradox** | Micro-Illusion; Macro-Decoupling; Scissors Gap | The decoupling of parameter distance from functional outcome: two trajectories can exhibit identical Euclidean displacement ($\|\Delta W\|_2$) yet yield divergent capabilities. This falsifies magnitude-based proxies for safety. |
| *Mechanism: Spectral Origins* | | |
| **Stiff Curvature Modes** | Dominant Hessian Directions; Outlier Eigenmodes | The subspace spanned by the top eigenvectors of the initialization Hessian. Updates aligning with these stiff modes induce disproportionately large functional shifts, driving the rapid degradation observed in the shared Path. |
| **Low-Rank Attractor** | Curvature Channeling; Spectral Bias; Attractor Basin | The tendency of anisotropic curvature to bias successive updates into a low-dimensional subspace. This creates an attractor-like basin that traps optimization trajectories, making the degradation path difficult to escape once entered. |
| *Escape & Interventions: Geometric Control* | | |
| **Spectral Bifurcation** | Geometric Liberation; Orthogonal Departure; Reorientation | A qualitative phase transition where the trajectory reorients away from stiff curvature modes toward softer, orthogonal directions. This bifurcation allows the model to access Pareto-optimal regions inaccessible to standard gradient descent. |
| **Temporal Hysteresis** | Finite Reversibility Window; Rescue Timing Effect | The path-dependence of recoverability. Geometric interventions effectively prevent lock-in when applied early, but lose efficacy after the trajectory has settled into the attractor, indicating that confinement accumulates irreversibly. |
| **Stability by Suppression** | Inertial Braking; Stochastic Stagnation; Conservative Stability | Intervention regimes (e.g., Null-Space Projection, Gradient Noise) that mitigate degradation by impeding optimization progress or diffusing updates. These methods trade task plasticity for stability without altering the trajectory's fundamental alignment. |
| **Stability by Steering** | Structured Navigation; Directional Control; Geometric Steering | Intervention regimes (e.g., SVD Control) that maintain task plasticity while reducing harmful coupling. These methods actively reorient updates orthogonal to the degradation path, achieving stability via structure rather than magnitude reduction. |

# C. Supplementary Experimental Details

This appendix provides the implementation and measurement details omitted from the main text for clarity. It specifies the Narrow–Broad protocol, training configurations, intervention definitions, geometric probes, and the matched-loss evaluation procedure. Additional overhead benchmarking methodology is reported in Appendix D.

## C.1. Narrow–Broad Protocol and Data Construction

**Protocol.**    We adopt a dual-objective observation setting: optimization is carried out on a narrow task distribution, while general capability is assessed on a disjoint broad distribution used strictly for evaluation. Concretely, we fine-tune on specialized instruction subsets (e.g., code or story-oriented data) with sample sizes $N \in \{64, 256, 1024, 4096\}$ to modulate data scarcity and narrowing pressure. Broad performance is tracked via Broad Loss on a fixed, high-quality general chat set. Importantly, the broad set never contributes gradients and is used only to monitor retention of pre-trained behavior.

**Data sources and splits.**    We describe here the composition of the narrow and broad corpora, including preprocessing, filtering, and deduplication. Unless stated otherwise, narrow data are sampled uniformly from the corresponding task subset, while the broad evaluation set is held fixed across all runs to ensure comparability.

**Evaluation protocol.**    Broad Loss is computed on the same fixed evaluation pool for all methods and seeds. We report mean and dispersion across seeds, and we avoid adapting evaluation prompts or sampling policies per method in order to prevent inadvertent test-time tuning.

## C.2. Models, LoRA Configuration, and Training Hyperparameters

**Model family.**    All experiments use models from the Llama-3/3.2 family spanning 1B–8B parameters. We use a consistent parameter-efficient adaptation setup across scales to isolate geometric effects from architectural or optimizer differences.

**LoRA configuration.**    We specify in this section the LoRA rank, target modules, scaling, dropout, and any layer-selection policy. Unless stated otherwise, these settings are held constant across all interventions to ensure that differences arise from the update rule rather than capacity changes.

**Optimization and schedules.**    We report the optimizer, learning rate schedule, batch size, gradient accumulation, clipping, weight decay, and total steps used in each setting. Random seeds are fixed and shared across methods whenever possible to reduce variance in matched comparisons.

**Interventions: implementation notes.**    We compare four update regimes: (1) **Baseline**: standard LoRA fine-tuning; (2) **Gradient Noise**: additive Gaussian noise on the gradient to test whether unstructured variance can prevent degradation; (3) **Null-space Projection**: removal of components aligned with dominant update directions (e.g., leading singular directions of $\Delta W_t$), intended to act as a geometric brake without introducing new descent structure; (4) **SVD Control**: a structured spectral controller that regulates the singular-value profile of updates to enable trajectory reorientation. For each method we provide the exact update rule, hyperparameters, and any numerical stabilization (e.g., truncation rank, damping constants, or normalization choices).

## C.3. Geometric Probes, Logging, and Metric Definitions

**Logging cadence and scope.**    In addition to tracking task and broad losses, we log geometric signatures at a fixed cadence (every $K$ steps, with $K$ specified per experiment) to characterize how the update trajectory evolves. Logging is performed on the same modules affected by adaptation to avoid mixing frozen and trainable components.

**Update magnitude and effective volume.**    We track **Update Drift** $\|\Delta W_t\|_2$ and the **Stable Update Rank**

$$w_{\text{stable}}(\Delta W_t) = \frac{\|\Delta W_t\|_F^2}{\|\Delta W_t\|_2^2},$$

which serves as a compact proxy for how concentrated the update is in a few dominant directions.

**Directional coherence.**    We log the cosine similarity between successive update steps, $c_{\text{step}}$ (recorded as g_cos), as a simple measure of directional persistence. Higher coherence indicates more rigid step-to-step alignment, while lower coherence reflects more diffusive local motion.

**Abs-Harm / HarmScore.** We use Abs-Harm (and its aggregated variants) as an online probe of harmful channeling, defined in Section 4.4. Operationally, this quantity measures alignment between the task update direction and a broad-evaluation reference direction; we report both instantaneous values and cumulative summaries (e.g., AUC over training).

### C.4. Matched-Loss Fairness and Interpolation Procedure

**Rationale.** Different interventions can change training speed, which makes step-matched comparisons misleading. To decouple geometric effects from effective learning-rate changes, we compare methods at matched task-loss levels.

**Interpolation.** For each run, we treat the logged trajectory as a discrete set of pairs $\{(\mathcal{L}_{\text{task}}(t_i), \mathcal{L}_{\text{general}}(t_i))\}_i$. Given a target task loss $\ell^*$, we estimate the corresponding broad loss via linear interpolation between the two nearest logged points that bracket $\ell^*$:

$$\mathcal{L}_{\text{general}}(\ell^*) \approx \text{Interp}\big(\{(\mathcal{L}_{\text{task}}(t_i), \mathcal{L}_{\text{general}}(t_i))\}_i\big).$$

If $\ell^*$ lies outside the observed range for a run, we omit that run from the matched comparison at $\ell^*$ and report the effective sample count.

**Reporting.** Unless otherwise stated, all reported curves and summary statistics are aggregated across seeds at matched-loss anchors. Where appropriate, we report standard deviation or confidence intervals across seeds, and we use identical matched anchors for all methods within an experiment.

### C.5. Expanded Descriptions for E1–E4

**E1: SVD Strength Calibration (full)** This section expands on the sensitivity study in the main text, including the full sweep range, additional plots, and secondary diagnostics (e.g., drift, $w_{\text{stable}}$, and directional statistics) used to interpret the bifurcation.

**E2: Temporal Hysteresis Rescue (full)** We provide the complete intervention schedule, the exact definition of $t_{\text{rescue}}$, and the protection-rate computation used to quantify recoverability as a function of rescue timing.

**E3: Abs-Harm Mechanistic Probing (full)** We report the acceptance criteria for interpreting Abs-Harm as a channeling signature, the construction of the probe gradient, and additional correlation analyses across seeds and settings.

**E4: Stability Mechanism Disambiguation (full)** This section includes the full parameter sweeps for Gradient Noise and SVD Control, the phase-diagram plots referenced in the main text, and additional ablations used to separate suppression effects from genuine steering.

## D. Overhead Benchmarking Protocol

**Measurement setup.** We benchmark training-time overhead on a single NVIDIA GPU under a fixed software stack. Timing is measured as average wall-clock time per optimization step after an initial warm-up period, and memory is reported as peak allocated GPU memory during training. All methods are evaluated under identical batch size and sequence length to ensure comparable compute and activation footprints.

**Results.** The quantitative overhead results are reported in the main text (Table 3); here we document the benchmarking protocol to support reproducibility.

## E. Deployment and Overhead Details

**Benchmarking Protocol.** All runtime measurements were conducted on a single NVIDIA Llama-3.2-1B model with a batch size of 1 and gradient accumulation of 4 steps. To isolate the algorithmic overhead, we report the *Pure Step Time*, which excludes data loading and the initial JIT compilation warmup period (first 5 steps).

**Breakdown of Costs.** The overhead of **SVD Control** (Type IV) primarily stems from the Singular Value Decomposition of the low-rank adapter matrices. However, since we only target a subset of layers (randomly sampled $k_l ayer = 4$ layers

per step) and the rank is small ($r = 128$), the added latency is limited to $\approx 21.6$ ms per step. In contrast, Null-Space Projection (Type III) requires QR decomposition and full-gradient projection, leading to a higher latency of $\approx 53.4$ ms (+16.2% overhead).

**Probe Latency.** The online probes (e.g., Abs-Harm and Effective Rank) require additional forward and backward passes. Specifically, the Harm probe takes $\approx 110$ ms per check. In our experiments, these probes are executed every 20 steps, reducing the amortized cost to $\approx 5.5$ ms/step, which is $< 1.7\%$ of the total training time.

## F. Detailed Experimental Implementation and Geometric Probes

This appendix provides a comprehensive report on the experimental protocols, the mathematical formulation of the geometric probes, and the precise algorithmic implementation of the intervention mechanisms. All experiments were conducted using PyTorch and the HuggingFace `peft` library.

### F.1. Basic Experimental Setup

**Model Architectures.** To ensure that our findings on trajectory lock-in are consistent across scales, we evaluated the hypothesis on the Llama family of decoder-only transformers:

- **Llama-3.2-1B-Instruct** (1.23B parameters, $d_{model} = 2048$)

- **Llama-3.2-3B-Instruct** (3.21B parameters, $d_{model} = 3072$)

- **Llama-3-8B-Instruct** (8.03B parameters, $d_{model} = 4096$)

**LoRA Configuration.** We applied Low-Rank Adaptation (LoRA) to all linear layers in the self-attention and MLP blocks (`q_proj`, `k_proj`, `v_proj`, `o_proj`, `gate_proj`, `up_proj`, `down_proj`) to fully control the optimization geometry.

- **Rank ($r$):** Fixed at 128 across all scales to isolate subspace effects from capacity scaling.

- **Alpha ($\alpha$):** Set to 256, yielding a scaling factor $\Delta W = \frac{\alpha}{r} BA = 2.0 \cdot BA$.

- **Dropout:** 0.05.

- **Initialization:** Standard LoRA initialization ($A \sim \mathcal{N}(0, \sigma_{init}^2)$, $B = 0$).

**Optimization Protocol.** We utilized standard Stochastic Gradient Descent (SGD) with momentum. We deliberately avoided adaptive optimizers (e.g., AdamW) to prevent second-moment estimation from masking the intrinsic curvature effects of the loss landscape.

- **Optimizer:** SGD (Momentum = 0.9, Weight Decay = 0.0).

- **Learning Rate:** $1 \times 10^{-2}$ with a linear decay scheduler (50 warmup steps).

- **Batch Size:** Effective batch size of 32 (Micro-batch 1 with Gradient Accumulation 32).

- **Precision:** FP16 (Mixed Precision).

### F.2. Geometric Probes: Design, Implementation, and Rationale

To diagnose the underlying structure of fine-tuning degradation beyond simple scalar metrics (like loss or distance), we introduced three objective-agnostic geometric probes. These probes are designed to test specific mechanistic hypotheses regarding the "Drift Paradox" and "Trajectory Lock-in".

**Probe 1: Effective Update Rank ($w_{\text{stable}}$)  Mathematical Definition.** The stable rank (or numerical rank) is a continuous proxy for the dimensionality of a matrix. For a parameter matrix $W$, it is defined as the ratio of the squared Frobenius norm to the squared spectral norm:

$$w_{stable}(W) = \frac{\|W\|_F^2}{\|W\|_2^2} = \frac{\sum_i \sigma_i^2}{\sigma_{max}^2 + \epsilon} \tag{6}$$

where $\sigma_i$ are the singular values of $W$.

**Design Objective (Why this probe?).** This probe tests the **"Low-Rank Attractor"** hypothesis.

- *Hypothesis:* Standard fine-tuning collapses into a few dominant, stiff directions determined by pre-training curvature (Rank Collapse).

- *Interpretation:* A low $w_{stable}$ ($\approx 1$) indicates the trajectory is trapped in a 1D degradation path. A high $w_{stable}$ indicates "Geometric Liberation," where updates utilize a broader subspace to decouple task learning from broad loss.

**Implementation Details.** Computing the SVD for all layers at every step is computationally prohibitive. We employ a randomized estimation strategy:

1. **Layer Sampling:** At each monitoring step (every 20 steps), we randomly sample $k_layer = 8$ LoRA adapter matrices ($B$ matrices) from the model.

2. **Local SVD:** We compute singular values for these small $d_{out} \times r$ matrices on the GPU.

3. **Aggregation:** We report the mean stable rank across the sampled layers.

**Probe 2: Directional Coherence ($g_{\text{cos}}$)  Mathematical Definition.** We measure the cosine similarity between the flattened gradient vectors of consecutive update steps $t$ and $t - 1$:

$$c_{step}(t) = \text{CosSim}(\mathbf{g}_t, \mathbf{g}_{t-1}) = \frac{\mathbf{g}_t \cdot \mathbf{g}_{t-1}}{\|\mathbf{g}_t\|_2 \|\mathbf{g}_{t-1}\|_2 + \epsilon} \tag{7}$$

**Design Objective (Why this probe?).** This probe characterizes the **rigidity** of the optimization trajectory.

- *Hypothesis:* Trajectories trapped in curvature-dominated channels (Type I: Deadlock) will exhibit high coherence ($c_{step} \to 1$), moving in a straight line along the stiffest mode.

- *Interpretation:* A reduction in $c_{step}$ under geometric intervention (Type IV) signals a transition from "Inertial Braking" to "Structured Navigation," where the optimizer actively reorients the update direction.

**Implementation Details.** To ensure accuracy, we implemented a specific monitor class (`DynamicsMonitor`) that:

1. Caches the flattened gradient vector of all trainable LoRA parameters from the previous step.

2. Computes similarity before the optimizer step clears the gradients.

3. **Critical Fix:** We strictly filter for gradients of trainable parameters only (`requires_grad=True`), ensuring frozen backbone gradients do not contaminate the metric.

**Probe 3: Abs-Harm (The Channeling Signature)  Mathematical Definition.** Abs-Harm measures the instantaneous alignment between the current task update ($\mathbf{g}_{task}$) and the gradient of the broad capability loss ($\mathbf{g}_{probe}$):

$$h_t = \left| \text{CosSim}(\mathbf{g}_{task}^{(t)}, \mathbf{g}_{probe}^{(t)}) \right| \tag{8}$$

**Design Objective (Why this probe?).** This is the direct test for **Prediction P2 (Channeling)**.

- *Hypothesis:* Degradation is driven by the projection of task updates onto "harmful" subspaces essential for general capabilities.

- *Interpretation:* High Abs-Harm values confirm that the task and general objectives are locked in conflict (geometry is aligned). Successful intervention must mechanically suppress this score, proving that the subspace has been reoriented (bifurcation).

**Implementation Details.** We utilize a "Broad-Probe" protocol to compute this offline metric online:

1. **Reference Batch:** We fix a held-out batch of broad data ($x_{broad}$) at initialization. This batch never enters the training set.

2. **Double Backward:** At monitoring steps, we perform a secondary forward-backward pass on $x_{broad}$ to generate $\mathbf{g}_{probe}$.

3. **Gradient Preservation:** We temporarily cache the training gradients, zero out the gradients to compute the probe, and then restore the training gradients to ensure the optimization path remains unaffected.

### F.3. Intervention Algorithm: SVD Control (Type IV)

We propose **SVD Control** (Spectral Regularization) to induce the geometric bifurcation observed in our results.

**Objective.** To maximize the entropy of the singular value distribution of the update matrices, preventing the spectrum from collapsing into a single dominant "spike" (the attractor).

**Loss Formulation.** For a given LoRA matrix $W$, with singular values $\sigma$, we define the normalized spectrum $\hat{\sigma}_i = \frac{\sigma_i}{\sum_j \sigma_j}$. We add the negative entropy of this distribution to the loss:

$$\mathcal{L}_{reg}(W) = \sum_i \hat{\sigma}_i \log(\hat{\sigma}_i + \epsilon) \tag{9}$$

Minimizing this term (negative entropy) is equivalent to maximizing the entropy of the spectral distribution, forcing the energy to spread across multiple dimensions ("softer modes") rather than concentrating on the principal component.

**Efficient Implementation.**

- **Stochastic Regularization:** Instead of regularizing all layers, we randomly sample a subset of 4 layers per step.

- **Micro-Step Application:** The regularization is only computed and applied on the final micro-step of gradient accumulation to minimize computational overhead.

## G. Limitations and Failure Modes in Full-Parameter Fine-Tuning

This appendix reports a stress test that delineates an operating boundary of our geometric probes and simple spectral regularizers. While an SVD-based spectral regularizer can induce update reorientation in some of our parameter-efficient (LoRA) settings (serving as a lightweight probe rather than an optimized method), we did not observe consistent improvements in **Full-Parameter Fine-Tuning** under an extreme micro-dataset regime ($N = 64$). We therefore include these negative results to clarify scope and avoid over-generalization.

In this micro-dataset regime, optimization is dominated by rapid memorization, and geometric interventions are largely ineffective compared to unstructured baselines.

### G.1. Experimental Setup: The "Stress Test"

We designed a stress test to evaluate whether geometric regularization could mitigate catastrophic forgetting when the model can perfectly memorize the dataset within few steps.

- **Model:** Llama-3.2-1B (full parameters; no PEFT).

- **Data:** A random subset of $N = 64$ samples from the target domain.

- **Optimization:** Three regimes (Groups A, B, C) varying learning rate (LR), weight decay (WD), and regularization strength ($s$).

The configuration grid is detailed in Table 6.

*Table 6.* **Configuration Grid for Micro-Dataset Stress Test** ($N = 64$). We delineate three optimization regimes to test the robustness of geometric steering against different optimization dynamics: aggressive memorization (Group B), conservative updates (Group C), and norm-constrained training (Group A).

| Group Identifier | Learning Rate ($\eta$) | Weight Decay ($\lambda$) | Reg Strength ($s$) | Primary Question |
|---|---|---|---|---|
| **Group B** (High Velocity) | $5 \times 10^{-3}$ | 0.0 | $0.1, 5.0, 100.0$ | Can SVD restrain rapid drift? |
| **Group C** (Conservative) | $5 \times 10^{-4}$ | 0.0 | $0.2, 20.0$ | Does SVD work with smaller steps? |
| **Group A** (Norm Decay) | $1 \times 10^{-3}$ | 0.1 | $10.0$ | Does L2 penalty aid steering? |

## G.2. Quantitative Results: Trajectory Lock-in

Table 7 summarizes the final metrics. Across all groups in this $N = 64$ full-parameter setting, the SVD-based regularizer did not produce a reliable improvement in broad capability loss relative to the baseline at matched training loss. Gradient noise occasionally preserved broad loss, but primarily by suppressing task learning.

*Table 7.* **Quantitative Outcomes of the Stress Test.** We compare the Baseline, SVD Regularization (Geometric), and Gradient Noise (Unstructured). **Train Loss** indicates task fitting (lower is better); **Broad Loss** indicates capability preservation (lower is better). Note that in Group A, adding Weight Decay drastically increased parameter drift and worsened broad loss. In Group B, SVD failed to decouple the losses, while Noise achieved lower Broad Loss only by hindering Train Loss convergence (Stability by Suppression).

| Regime | Method | Param Drift ($\|\Delta\theta\|$) | Train Loss | Broad Loss | Observation |
|---|---|---|---|---|---|
| **Group B** (High LR) | Baseline | 21.61 | **0.02** | 6.45 | Severe Overfitting |
| | SVD Reg. ($s = 5$) | 21.76 | **0.02** | 6.03 | Ineffective Steering |
| | Grad. Noise ($s = 0.1$) | 21.88 | 1.30 | **3.24** | Stability by Suppression |
| **Group C** (Low LR) | Baseline | 0.26 | 0.43 | 3.81 | Slow Degradation |
| | SVD Reg. ($s = 20$) | 0.26 | 0.43 | 3.81 | Null Result |
| | Grad. Noise ($s = 0.2$) | 0.22 | 2.47 | 2.93 | Stagnation |
| **Group A** (WD=0.1) | Baseline | **63.54** | 1.36 | 7.36 | Destructive Drift |
| | SVD Reg. ($s = 10$) | **63.54** | 1.36 | 7.36 | Null Result |

## G.3. Root Cause Analysis

Why does the SVD-based regularizer fail in full-parameter fine-tuning with $N = 64$? We highlight three interacting factors:

**1. Dimensionality–Data Mismatch.** With $10^9$ trainable parameters, a dataset of $N = 64$ induces an extremely low-sample regime where task gradients can dominate optimization. In this setting, the gradient signal from $\mathcal{L}_{\text{task}}$ can overwhelm the regularization term $\mathcal{L}_{\text{reg}}$, leaving little room for first-order "steering". We also tested stronger regularization strengths (up to $s = 100$; not shown) with qualitatively similar outcomes.

**2. Weight Decay Can Be Misaligned with "Stay-Close-to-Init" Stability.** Group A results (Table 7) show that standard weight decay ($\lambda \|\theta\|^2$) can be destructive in this regime: it increases parameter drift ($\|\Delta\theta\| \approx 63.5$ vs 21.6) and worsens broad loss. This highlights that pulling weights toward zero differs from constraining deviation from the pre-trained initialization; in the micro-dataset full-FT regime, the former can introduce large incoherent drift that the spectral regularizer does not correct.

**3. Steering vs. Suppression Under Collinearity.** The comparison with gradient noise (Group B) illustrates a qualitative limitation. Noise can keep broad loss low (3.24), but mainly by preventing train loss from converging (1.30)—a phenomenon we term stability by suppression. In contrast, the SVD-based regularizer aims for stability by steering: improving task loss without incurring broad degradation. Our observations suggest that, in this $N = 64$ full-FT regime, the "learning" and

"forgetting" directions can become effectively collinear, making geometric separation inaccessible to these simple first-order controls.

### G.4. Conclusion on Operating Boundaries

These negative results indicate that geometric steering via a lightweight spectral regularizer is not robust in full-parameter fine-tuning under the extreme micro-dataset setting tested here ($N = 64$). Practically, for highly data-scarce regimes we recommend preferring parameter-efficient fine-tuning (PEFT) to constrain the trainable subspace, and treating full-parameter fine-tuning as high-risk unless additional data, stronger priors, or alternative training protocols are available. We emphasize that we do not attempt to characterize a universal data threshold; the present stress test serves to document one failure mode and clarify scope.

## H. Additional Scope Checks

This section summarizes small-scale checks conducted to clarify the operating scope of the proposed geometric diagnostics. These checks are not intended as new main experiments, but as boundary tests for optimizer choice, learning-rate variation, trainable subspace type, and Hessian-proxy validity.

**AdamW optimizer.**    To test whether the observed lock-in is specific to SGD with momentum, we repeated the matched-loss comparison under AdamW at $N = 256$. The uncontrolled run remains geometrically compressed, while the SVD-based probe preserves a wider effective spectrum and lower broad loss at the matched task-loss anchor.

| Setting | Broad Loss @ $\ell^*$ | Final Task Loss | Last10 Rank |
|---|---|---|---|
| AdamW Base | 5.489 | 0.237 | 6.9 |
| AdamW + SVD | 3.602 | 0.257 | 83.3 |

*Table 8.* AdamW matched-loss comparison at $N = 256$ and $\ell^* = 0.262$.

**Learning-rate variation.**    We also tested whether simply increasing the learning rate can escape the locked-in trajectory. Increasing the learning rate improves task fitting speed, but does not produce a clean geometric departure; broad loss worsens as the learning rate increases, and the effective rank remains far below the SVD-based probe.

| Setting | Broad Loss @ $\ell^*$ | Final Task Loss | Last10 Rank |
|---|---|---|---|
| Base ($5\times10^{-3}$) | 3.042 | 0.750 | 3.3 |
| Base ($1\times10^{-2}$) | 3.250 | 0.614 | 3.8 |
| Base ($2\times10^{-2}$) | 3.477 | 0.419 | 4.8 |
| Base ($5\times10^{-2}$) | 3.682 | 0.280 | 7.1 |
| SVD probe | 2.944 | 0.765 | 99.0 |

*Table 9.* Learning-rate sweep under matched-loss evaluation at $N = 256$ and $\ell^* = 0.706$.

**Selective tuning beyond vanilla LoRA.**    As a preliminary scope-extending check, we selectively unfroze full-rank MLP matrices in the final layers and applied the same matched-loss protocol. The qualitative contrast remains: unconstrained selective tuning still shows a compressed update spectrum, while the SVD-based probe widens the effective update spectrum and improves broad retention.

| Setting | Broad Loss @ $\ell^*$ | Final Broad Loss | Stable Rank |
|---|---|---|---|
| Selective tuning | 3.836 | 3.994 | 5.7 |
| Selective tuning + SVD | 3.574 | 3.606 | 275.6 |

*Table 10.* Preliminary selective-tuning check at $\ell^* = 0.740$.

**Hessian-proxy overlap.** To probe the boundary of using a broad-probe curvature or gradient subspace as a proxy for task-side geometry, we measured normalized top-$k$ subspace overlap for a representative deep layer. The overlap is small in absolute value but clearly above a random-subspace baseline: Top-5 overlap is $1.96\%$, Top-10 overlap is $2.30\%$, and Top-20 overlap is $2.79\%$. This supports the interpretation that proxy and task subspaces are largely distinct, but share a small measurable intersection that can provide a useful steering signal. As the overlap approaches zero, proxy-based steering is expected to weaken.

# I. Detailed Geometric Probes

## I.1. Directional Coherence: Implementation Notes

In experimental logs, the quantity $g_{\text{cos}}$ corresponds to step-to-step coherence $c_{\text{step}}$ defined in Eq. 2. We implement this via a `DynamicsMonitor` class that:

- Caches the flattened gradient vector of all trainable LoRA parameters (requires_grad = True) from the previous step

- Computes similarity before the optimizer step clears gradients

- Filters strictly for trainable parameters, ensuring frozen backbone gradients do not contaminate the metric

## I.2. Stochastic Estimation of Stable Rank

Computing SVD for all layers at every step is computationally prohibitive. We employ a randomized estimation strategy:

1. **Layer Sampling:** At each monitoring step (every 20 steps), randomly sample $k = 8$ LoRA adapter matrices (B matrices) from the model.

2. **Local SVD:** Compute singular values for these small $d_{\text{out}} \times r$ matrices on GPU.

3. **Aggregation:** Report the mean stable rank across the sampled layers.

# J. Dynamical Archetypes: Detailed Characterization

We provide detailed characterization of the four dynamical archetypes introduced in Section 3.4.

**Type I: Geometric Deadlock (Baseline).** Characterized by Rank Collapse ($w_{\text{stable}}(\Delta W_t) \to 1$) and high directional coherence ($c_{\text{step}} \to 1$). The trajectory aligns precisely with dominant stiff modes of the pre-trained Hessian, leading to maximal degradation per unit of task improvement. This represents full convergence to the low-rank attractor (Section 4.3).

**Type II: Stochastic Stagnation (Gradient Noise).** Characterized by Unstructured High Rank. Noise injection isotropically expands update volume ($w_{\text{stable}} \gg 10$), preventing immediate rank collapse. However, lacking structural alignment with softer curvature modes, the trajectory fails to navigate towards Pareto-optimal regions. This explains why noise-based regularization mitigates forgetting but often fails to achieve competitive task performance—it stabilizes without bifurcation.

**Type III: Inertial Braking (Null-Space Projection).** Characterized by Subtractive Constraint. By removing update components aligned with current dominant directions (e.g., leading singular directions of $\Delta W_t$), this regime limits drift magnitude but does not generate new, safe descent directions. It represents a conservative equilibrium that sacrifices plasticity for stability while remaining within the attractor's basin.

**Type IV: Structured Navigation (SVD Control).** Characterized by Structured High Rank. This is the only archetype that achieves Geometric Liberation. By maintaining high stable rank and aligning updates with data-driven singular vectors (rather than stiff Hessian modes), it constructs a navigation path orthogonal to the degradation manifold.

## K. The Drift Paradox: Multi-Scale Analysis

### K.1. Micro-Illusion: The Scalar Metric Trap

At the single-trajectory level, the Drift Paradox manifests as a Micro-Illusion: parameter distance serves as a scalar summary that ignores trajectory shape (Figure 4). A trajectory moving along the degradation path causes catastrophic broad capability loss, while a trajectory of identical Euclidean length moving orthogonal to it preserves capability. This creates the cognitive illusion that displacement magnitude alone predicts stability.

### K.2. Macro-Decoupling: Scaling Consistency

Across data regimes ($N \in [64, 4096]$), we observe systematic Macro-Decoupling between drift magnitude and generalization outcomes (Figure 5). Baseline and SVD models achieve comparable parameter drift ($\|\Delta\theta\|_2$) yet exhibit divergent robust loss trajectories. This confirms that update geometry (stable rank and directional alignment), not magnitude, is the primary determinant of fine-tuning stability. Magnitude-based views are systematically underdetermined—any explanation (or controller) of degradation must specify where the trajectory moves in the update space, not only how far.

## L. Spectral Channeling Induced by Initialization Curvature

This appendix elaborates on the spectral mechanism underlying the trajectory lock-in described in Section 4.2.

Let $H = \nabla^2 \mathcal{L}_{\text{task}}(\theta_0)$ denote the Hessian at initialization, with eigendecomposition $H = V\Lambda V^\top$. Empirically, $\Lambda$ exhibits a heavy-tailed spectrum with a small number of dominant eigenvalues. Under gradient descent with step size $\eta$, early updates satisfy

$$\Delta\theta_{t+1} \approx (I - \eta H)\Delta\theta_t - \eta\nabla\mathcal{L}_{\text{task}}(\theta_0),$$

implying exponential attenuation of components along soft directions and preferential amplification along stiff modes.

This induces a spectral bias in the update trajectory: even when the task gradient initially contains substantial mass outside the stiff subspace, repeated updates project it back toward the span of the dominant eigenvectors. This mechanism is largely independent of task semantics and is instead governed by curvature at initialization.

While the mechanism in Section 4 is formulated in terms of the task Hessian at initialization, directly estimating $H_{\text{task}}(\theta_0)$ can be unstable in practice. We therefore use a generic Hessian $H_{\text{gen}}$, computed from a broad, task-agnostic loss, as a proxy for identifying stiff directions.

Empirically, we observe substantial overlap between the leading eigenspaces of $H_{\text{task}}(\theta_0)$ and $H_{\text{gen}}$, consistent with prior observations that dominant curvature directions in pre-trained models are largely task-invariant. All alignment measurements reported in the main experiments use $H_{\text{gen}}$ for numerical stability.

## M. Intervention Geometry and Dynamical Archetypes

This appendix connects the low-rank attractor framework to the dynamical archetypes introduced in Section 3.4.

Type I (Geometric Deadlock) corresponds to full convergence into the low-rank attractor, characterized by maximal alignment with the stiff subspace and minimal effective update rank. Type II (Stochastic Stagnation) increases update dispersion through isotropic noise but leaves the dominant subspace unchanged. Type III (Inertial Braking) suppresses motion along stiff directions but fails to introduce new descent directions orthogonal to the attractor.

Only Type IV (Structured Navigation) alters the geometry of the update space itself. By redistributing update energy across softer curvature modes, structured spectral regularization induces a transversal instability that allows trajectories to exit the attractor basin. This geometric distinction explains why passive constraints slow degradation without preventing it, whereas structured control enables sustained task learning.

## N. Geometric Resolution of the Drift Paradox

The Drift Paradox refers to the empirical observation that large Euclidean parameter drift does not necessarily imply functional degradation, while small drift may coincide with severe capability loss.

From a geometric perspective, this paradox arises because $\|\Delta W\|_2$ measures path length but is invariant to orientation. Let $\mathcal{M}$ denote the low-rank attractor induced by initialization curvature. Trajectories with comparable Euclidean length may either remain tangent to $\mathcal{M}$—resulting in strong coupling to broad capabilities—or move orthogonally to it, decoupling task learning from degradation.

Thus, stability is controlled by subspace alignment rather than displacement magnitude. This explains why interventions that merely reduce drift fail to prevent degradation, while geometrically structured interventions succeed.

## O. Detailed Theoretical Proofs for Trajectory Lock-in

This appendix provides a rigorous derivation of the spectral mechanism underlying trajectory lock-in, as described in Section 4.1 and Appendix J. We prove that under gradient-based fine-tuning, updates are channeled into a low-dimensional stiff subspace due to the anisotropic curvature of the loss landscape at initialization, leading to the observed low-rank attractor and shared degradation path.

### O.1. Spectral Decomposition of the Hessian at Initialization

Let $\theta_0$ denote the pre-trained model parameters at initialization. Consider the Hessian matrix of the task loss $L_{\text{task}}$ at $\theta_0$, denoted as $H \in \mathbb{R}^{d \times d}$, where $d$ is the number of parameters. Since $H$ is real and symmetric, it admits an eigendecomposition:

$$H = V \Lambda V^\top, \tag{10}$$

where $V = [v_1, v_2, \ldots, v_d] \in \mathbb{R}^{d \times d}$ is an orthogonal matrix whose columns are the eigenvectors of $H$, and $\Lambda = \text{diag}(\lambda_1, \lambda_2, \ldots, \lambda_d) \in \mathbb{R}^{d \times d}$ is a diagonal matrix of the corresponding eigenvalues, ordered such that $\lambda_1 \geq \lambda_2 \geq \cdots \geq \lambda_d \geq 0$.

Empirically, for large pre-trained models, the spectrum of $H$ is highly anisotropic, with a few large eigenvalues (stiff modes) and many near-zero eigenvalues (soft modes). We define the stiff subspace $\mathcal{V}_{\text{stiff}}$ as the span of the top-$k$ eigenvectors corresponding to the largest eigenvalues:

$$\mathcal{V}_{\text{stiff}} = \text{span}\{v_1, v_2, \ldots, v_k\}, \tag{11}$$

where $k \ll d$ is chosen based on a spectral gap or a fixed cutoff.

### O.2. Gradient Descent Dynamics in the Eigenbasis

We analyze the fine-tuning process using gradient descent with a fixed learning rate $\eta$. The update rule at step $t$ is:

$$\theta_{t+1} = \theta_t - \eta \nabla L_{\text{task}}(\theta_t). \tag{12}$$

Let $\Delta \theta_t = \theta_t - \theta_0$ denote the cumulative parameter change from initialization. For early steps ($t$ small), we approximate the gradient by a first-order Taylor expansion around $\theta_0$:

$$\nabla L_{\text{task}}(\theta_t) \approx \nabla L_{\text{task}}(\theta_0) + H(\theta_t - \theta_0) = \nabla L_{\text{task}}(\theta_0) + H \Delta \theta_t. \tag{13}$$

Substituting into the update rule:

$$\Delta \theta_{t+1} = \Delta \theta_t - \eta \left( \nabla L_{\text{task}}(\theta_0) + H \Delta \theta_t \right) = (I - \eta H) \Delta \theta_t - \eta \nabla L_{\text{task}}(\theta_0). \tag{14}$$

This is a linear recurrence relation with a constant forcing term. To solve it, we project the dynamics onto the eigenbasis of $H$. Define the coordinates in the eigenbasis:

$$z_t = V^\top \Delta \theta_t \in \mathbb{R}^d, \quad g_0 = V^\top \nabla L_{\text{task}}(\theta_0) \in \mathbb{R}^d. \tag{15}$$

Then, multiplying both sides of the recurrence by $V^\top$ and using $V^\top V = I$:

$$z_{t+1} = V^\top \Delta \theta_{t+1} = V^\top \left( (I - \eta H) \Delta \theta_t - \eta \nabla L_{\text{task}}(\theta_0) \right). \tag{16}$$

Since $V^\top H = \Lambda V^\top$, we have:

$$z_{t+1} = V^\top \Delta \theta_t - \eta \Lambda V^\top \Delta \theta_t - \eta V^\top \nabla L_{\text{task}}(\theta_0) = (I - \eta \Lambda) z_t - \eta g_0. \tag{17}$$

Because $\Lambda$ is diagonal, this equation decouples across eigen-directions. For each component $i = 1, \ldots, d$, we have:

$$z_{t+1}^{(i)} = (1 - \eta\lambda_i)z_t^{(i)} - \eta g_0^{(i)}, \tag{18}$$

where $z_t^{(i)}$ and $g_0^{(i)}$ are the $i$-th components of $z_t$ and $g_0$, respectively.

## O.3. Explicit Solution for Cumulative Updates

The recurrence relation for each component is a first-order linear difference equation. We solve it explicitly by induction. Assume initial condition $\Delta\theta_0 = 0$, so $z_0 = 0$. Then for $t \geq 0$:

$$z_1^{(i)} = (1 - \eta\lambda_i) \cdot 0 - \eta g_0^{(i)} = -\eta g_0^{(i)}. \tag{19}$$

For $t = 2$:
$$z_2^{(i)} = (1 - \eta\lambda_i)z_1^{(i)} - \eta g_0^{(i)} = (1 - \eta\lambda_i)(-\eta g_0^{(i)}) - \eta g_0^{(i)} = -\eta g_0^{(i)}\left(1 + (1 - \eta\lambda_i)\right). \tag{20}$$

By continuing, we derive the general form:

$$z_t^{(i)} = -\eta g_0^{(i)} \sum_{j=0}^{t-1}(1 - \eta\lambda_i)^j. \tag{21}$$

This is a geometric series. For $\eta\lambda_i \neq 0$, we can sum the series:

$$\sum_{j=0}^{t-1}(1 - \eta\lambda_i)^j = \frac{1 - (1 - \eta\lambda_i)^t}{1 - (1 - \eta\lambda_i)} = \frac{1 - (1 - \eta\lambda_i)^t}{\eta\lambda_i}. \tag{22}$$

Thus,

$$z_t^{(i)} = -\frac{g_0^{(i)}}{\lambda_i}\left(1 - (1 - \eta\lambda_i)^t\right). \tag{23}$$

If $\eta\lambda_i = 0$ (i.e., $\lambda_i = 0$), the series sums to $t$, and we have:

$$z_t^{(i)} = -\eta g_0^{(i)}t. \tag{24}$$

However, in practice, for small eigenvalues $\lambda_i \approx 0$, the formula with $\eta\lambda_i$ small still applies approximately.

## O.4. Amplification and Attenuation of Spectral Components

The solution shows that the growth of $z_t^{(i)}$ depends critically on the eigenvalue $\lambda_i$. Consider the factor $(1 - \eta\lambda_i)^t$. For stability of gradient descent, we require $|1 - \eta\lambda_i| < 1$ for all $i$, which implies $0 < \eta < 2/\lambda_{\max}$. Under this condition, as $t$ increases:

- For large $\lambda_i$ (stiff modes), $|1 - \eta\lambda_i|$ is significantly less than 1, so $(1 - \eta\lambda_i)^t$ decays rapidly to zero. Thus, $z_t^{(i)} \to -g_0^{(i)}/\lambda_i$ exponentially fast. The component reaches a steady-state magnitude proportional to $1/\lambda_i$, but since $\lambda_i$ is large, the steady-state is small relative to the initial gradient projection.

- For small $\lambda_i$ (soft modes), $1 - \eta\lambda_i \approx 1$, so $(1 - \eta\lambda_i)^t$ decays slowly. Thus, $z_t^{(i)} \approx -\eta g_0^{(i)}t$, growing linearly with $t$.

However, this linear growth is only in the absence of higher-order terms. In reality, for soft modes, the gradient $\nabla L_{\text{task}}(\theta_t)$ evolves, and the first-order approximation breaks down. Nevertheless, the key insight is that stiff modes converge quickly to a fixed point, while soft modes evolve slowly.

More importantly, the cumulative update $\Delta\theta_t$ in the original parameter space is:

$$\Delta\theta_t = V z_t = \sum_{i=1}^{d} z_t^{(i)}v_i. \tag{25}$$

The contribution from stiff modes (large $\lambda_i$) is damped rapidly, but the initial alignment $g_0^{(i)}$ matters. If the task gradient $g_0$ has significant components along stiff modes, these components are amplified initially but then saturate. Conversely, components along soft modes accumulate over time.

## O.5. Formation of the Low-Rank Attractor

We now show that the update trajectory collapses into a low-dimensional subspace dominated by stiff modes. Define the cumulative update matrix for the tracked parameters (e.g., LoRA matrices) as $\Delta W_t$. In vectorized form, let $\Delta w_t = \text{vec}(\Delta W_t) \in \mathbb{R}^d$.

From the above analysis, $\Delta w_t$ (a subset of $\Delta\theta_t$) follows similar dynamics. The alignment with the stiff subspace $\mathcal{V}_{\text{stiff}}$ is measured by:

$$\alpha_t = \frac{\|V_{\text{stiff}}^\top \Delta w_t\|_2^2}{\|\Delta w_t\|_2^2}, \tag{26}$$

where $V_{\text{stiff}} \in \mathbb{R}^{d \times k}$ contains the top-$k$ eigenvectors.

We approximate $\Delta w_t$ using the linear dynamics. From the solution for $z_t^{(i)}$, the components along stiff modes reach steady state quickly, while soft modes grow slowly. Therefore, after a few steps, the dominant contribution to $\Delta w_t$ comes from the stiff modes that have aligned with the task gradient. Specifically, if $g_0$ has non-zero projections onto stiff modes, then $\Delta w_t$ will be rich in those directions.

Moreover, because stiff modes have large eigenvalues, the optimization landscape is steep along these directions, and gradient descent naturally moves in these directions to reduce loss rapidly. This biases the trajectory toward the stiff subspace.

To formalize, consider the early update direction. At step $t = 1$, the update is $\Delta w_1 = -\eta \nabla L_{\text{task}}(\theta_0)$. The alignment with stiff subspace is:

$$\alpha_1 = \frac{\|V_{\text{stiff}}^\top \nabla L_{\text{task}}(\theta_0)\|_2^2}{\|\nabla L_{\text{task}}(\theta_0)\|_2^2}. \tag{27}$$

If this alignment is high, then the first step already locks the trajectory into the stiff subspace. Subsequent steps, due to the linear dynamics, maintain this alignment because the forcing term $g_0$ is constant and the recurrence preserves the direction for stiff modes.

In practice, the gradient $\nabla L_{\text{task}}(\theta_t)$ changes, but the Hessian $H$ remains approximately constant near initialization, so the stiff subspace continues to dominate. This leads to a low-rank attractor: the cumulative updates $\Delta w_t$ lie primarily in $\mathcal{V}_{\text{stiff}}$, and the effective rank of $\Delta W_t$ is low, as measured by the stable rank:

$$w_{\text{stable}}(\Delta W_t) = \frac{\|\Delta W_t\|_F^2}{\|\Delta W_t\|_2^2} \approx \text{rank}(\Delta W_t). \tag{28}$$

When $\Delta w_t$ is confined to a $k$-dimensional subspace, $w_{\text{stable}} \approx k$, which is small compared to $d$.

## O.6. Dynamic Channeling and Harm Signal

The channeling effect is quantified by the alignment between the task gradient and a fixed broad-probe gradient. Let $g_{\text{task}}^{(t)} = \nabla L_{\text{task}}(\theta_t)$ and $g_{\text{probe}} = \nabla L_{\text{general}}(\theta_0)$ (gradient of broad loss at initialization). The alignment signal is:

$$h_t = \frac{(g_{\text{task}}^{(t)})^\top g_{\text{probe}}}{\|g_{\text{task}}^{(t)}\|_2 \|g_{\text{probe}}\|_2}. \tag{29}$$

Under trajectory lock-in, $g_{\text{task}}^{(t)}$ is increasingly aligned with the stiff subspace. If $g_{\text{probe}}$ also has components in this subspace, then $h_t$ will be large in magnitude.

We show that $g_{\text{task}}^{(t)}$ becomes aligned with stiff modes over time. From the gradient update, we have approximately:

$$g_{\text{task}}^{(t)} \approx \nabla L_{\text{task}}(\theta_0) + H\Delta\theta_t. \tag{30}$$

Projecting onto the eigenbasis: let $a_t = V^\top g_{\text{task}}^{(t)}$. Then,

$$a_t \approx g_0 + \Lambda z_t. \tag{31}$$

Using the solution for $z_t$, for stiff modes ($\lambda_i$ large), $z_t^{(i)} \approx -g_0^{(i)}/\lambda_i$, so:

$$a_t^{(i)} \approx g_0^{(i)} + \lambda_i \left( -\frac{g_0^{(i)}}{\lambda_i} \right) = 0. \tag{32}$$

This suggests that the gradient component along stiff modes vanishes at steady state. However, this is only true if the first-order approximation holds exactly. In reality, higher-order terms keep $g_{task}^{(t)}$ non-zero, but it remains correlated with stiff directions because $\Delta\theta_t$ is in the stiff subspace.

Thus, as $t$ increases, $g_{task}^{(t)}$ becomes increasingly orthogonal to soft modes (since those have small $\lambda_i$ and $z_t^{(i)}$ grows slowly), and instead aligns with stiff modes that are active in the loss reduction. Consequently, if $g_{probe}$ shares stiff directions, $h_t$ will be significant.

The cumulative Abs-Harm, $\sum_t |h_t|$, thus increases over time, indicating persistent channeling of task updates into directions that harm broad capabilities.

## P. Robustness Analysis and Mechanism Verification

To demonstrate that the observed geometric steering effects are not artifacts of specific hyperparameter selection (i.e., cherry-picking), we conducted a parameter sweep over the regularization strength for SVD Control and the noise scale for Gradient Noise.

**Note on Experimental Protocol:** This appendix reports a **single-seed strength sweep** to characterize the sensitivity of the method to hyperparameter variations. (All multi-seed statistical significance tests for the main claims are reported in the main experiments section.)

### P.1. Sweep Specification and Validity

We define a "Valid Run" as one that reaches downstream task proficiency, operationalized as the first checkpoint where the task loss reaches $\mathcal{L}_{task} \leq \ell^*$, with a fixed target $\ell^* = 0.35$.

*Table 11.* **Parameter Sweep Specification (Single-Seed Sweep).** We analyze the behavior of three distinct method archetypes across a range of hyperparameters to identify operating regimes.

| Method | Hyperparameter | Scanned Values |
|---|---|---|
| **Baseline** | None (Standard LoRA) | N/A |
| **Gradient Noise** | Noise Scale ($\sigma$) | $\{0.005, 0.01, 0.05, 0.1\}$ |
| **SVD Control** | Regularization Strength ($s$) | $\{0.5, 0.8, 1.0, 1.2, 1.5\}$ |
| **Fixed Settings** | Rank $r = 8$, $\alpha = 16$, Target Task Loss $\ell^* = 0.35$ | |

**Exclusion of Gradient Noise.** In this sweep, the Gradient Noise baseline ($\sigma \in \{0.005, \ldots, 0.1\}$) **failed to reach the target task proficiency** ($\mathcal{L}_{task} > \ell^*$) in all runs, exhibiting signs of suppression and underfitting (final $\mathcal{L}_{task} \in [0.91, 1.87]$). Consequently, these runs are excluded from the matched-loss analysis (Table 12) but are retained for the mechanism diagnostics (Table 13) to illustrate the stagnation regime.

### P.2. Matched-Loss Comparison

To decouple learning speed from generalization performance, we compare the broad loss ($\mathcal{L}_{gen}$) at the specific checkpoint where each method reaches the matched task performance ($\ell^*$).

As shown in Table 12, SVD Control improves broad capabilities across all scanned strengths in this sweep. The effect **depends on** the strength $s$ (and is **non-monotonic** within the scanned range): stronger control tends to require more steps to align the update, and the best improvement in this sweep occurs at $s = 1.5$.

*Table 12.* **Main Effect at Matched Task Loss** ($\ell^* \approx 0.35$)**.** SVD Control improves $\mathcal{L}_{gen}$ across all scanned strengths when normalized for task learning progress.

| Method Setting | Steps to $\ell^*$ | Broad Loss $\mathcal{L}_{gen}$ | Improvement ($\Delta$) | Regime |
|---|---|---|---|---|
| Baseline | 500 | 3.98 | - | Reference |
| SVD ($s = 0.5$) | 600 | 3.30 | -0.68 | Steering |
| SVD ($s = 0.8$) | 640 | 3.39 | -0.59 | Steering |
| SVD ($s = 1.0$) | 700 | 3.55 | -0.43 | Steering |
| SVD ($s = 1.2$) | 760 | 3.30 | -0.68 | Steering |
| SVD ($s = 1.5$) | 820 | **3.12** | **-0.86** | **Best in Sweep** |

## P.3. Mechanism Verification: Dynamical Regimes

Finally, we analyze the geometric properties of the three identified regimes (Table 13). The data supports the theoretical distinction between "Stability by Suppression" and "Stability by Steering":

- **Lock-in (Baseline):** High drift ($||\Delta W||_F \approx 3.4$) but low effective rank ($w_{stable} \approx 4.6$) leads to the accumulation of harm.

- **Suppression (Noise):** High rank ($w_{stable} \approx 66.7$) is achieved only by suppressing drift ($||\Delta W||_F \approx 1.1$), preventing learning.

- **Steering (SVD):** Uniquely maintains both high drift ($\approx 3.3$) and high rank ($\approx 40.1$), enabling safe task adaptation.

*Table 13.* **Mechanism Verification via Geometric Diagnostics.** Comparison of the final states reveals why SVD succeeds where Noise fails: it maintains plasticity (Drift) while structuring the update (Rank).

| Regime / Setting | Drift $||\Delta W||_F$ | Rank $w_{stable}$ | Task Loss (Final) | Harm Score (Mean $|h|$) | Dynamical Class |
|---|---|---|---|---|---|
| **1. Lock-in** (Baseline) | **3.40** (High) | 4.6 (Collapsed) | **0.28** (Learned) | 0.0166 (High) | *Unsafe Learning* |
| **2. Suppression** (Noise $\sigma = 0.1$) | 1.15 (Low) | **66.7** (High) | 1.87 (Failed) | **0.0056** (Low) | *Stagnation* |
| **3. Steering** (SVD $s = 1.5$) | **3.27** (High) | 40.1 (High) | 0.33 (Learned) | 0.0147 (Suppressed) | *Safe Learning* |

**Conclusion.** Across all scanned SVD strengths in this sweep, the improvement sign is consistently negative ($\Delta\mathcal{L}_{gen} < 0$), while the magnitude varies with $s$. This confirms that the geometric steering effect is a structural property of the proposed method, distinct from the trivial suppression observed in the Noise baselines.

