# OpenReview forum: "The Geometry of Narrow Fine-Tuning Degradation: Trajectory Lock-in and Spectral Bifurcation"
_ICML.cc/2026/Conference — ICML 2026 regular_

### Official Review · Reviewer_837X · 2026-03-06

**Soundness:** 2
**Presentation:** 2
**Significance:** 4
**Originality:** 3
**Overall Recommendation:** 4
**Confidence:** 4

**Summary:**

This paper analyzes broad capability degradation during narrow-task LoRA fine-tuning through the lens of parameter-space geometry. The authors identify "trajectory lock-in," an empirical phenomenon where diverse interventions collapse onto a shared degradation curve. To diagnose this, they introduce geometric probes (update drift, stable rank, directional coherence) and an online Abs-Harm metric, exposing a "Drift Paradox" where identical parameter displacement yields divergent capability outcomes. Finally, they demonstrate that an SVD-based spectral regularizer can induce a "spectral bifurcation" to redirect updates toward softer curvature modes, though it notably fails in full-parameter micro-dataset regimes.

**Compliance With Llm Reviewing Policy:**

Affirmed.

**Final Justification:**

Thank you for the detailed rebuttal. The additional experiments and clarifications addressed my main concerns, especially around optimizer dependence, the learning-rate question, and the practical role of the probes, so I am comfortable moving my score to **4 (weak accept)**. My only remaining reservation is that the novelty claim is still not fully clean to me: the paper’s central message seems close to existing ideas that update direction matters more than magnitude and that flatter/softer directions help generalization, so I would still encourage the authors to state more explicitly what the paper adds beyond that prior literature and to sharpen the contribution in the final version.

**Key Questions For Authors:**

1. How do these lock-in and steering phenomena hold up when evaluated with adaptive optimizers like AdamW?
2. How does SVD Control compare to recent steering methods like OPLoRA, MiLoRA, and CLoRA [3, 4, 5]? Do these baselines achieve true "spectral bifurcation," or are they trapped in "inertial braking" within your geometric framework?
3. The current paper shows LoRA success and one extreme full-FT failure for SVD control, but not where the breakdown begins. Can you ablate across trainable subspace size and data regime to identify this?
4. Beyond theoretical diagnostics, what are the immediate practical advantages of the proposed geometric probes? Can the online Abs-Harm metric be utilized dynamically as an automated early-stopping criterion?


*References*


[1] Aghajanyan, A., et al. "Intrinsic Dimensionality Explains the Effectiveness of Language Model Fine-Tuning." *Proceedings of the Association for Computational Linguistics (ACL)*, 2021.

[2] Kirkpatrick, J., et al. "Overcoming Catastrophic Forgetting in Neural Networks." *Proceedings of the National Academy of Sciences (PNAS)*, 2017.

[3] Xiong, Y., et al. "OPLoRA: Orthogonal Projection LoRA Prevents Catastrophic Forgetting during Parameter-Efficient Fine-Tuning." *arXiv preprint*, 2025.

[4] Wang, X., et al. "MiLoRA: Harnessing Minor Singular Components for Parameter-Efficient LLM Finetuning." *Proceedings of the North American Chapter of the Association for Computational Linguistics (NAACL)*, 2025.

[5] Zhang, X., et al. “C-LoRA: Continual Low-Rank Adaptation for Pre-trained Models.” *arXiv preprint*, 2025.

[6] Keskar, N. S., et al. "On Large-Batch Training for Deep Learning: Generalization Gap and Sharp Minima." *International Conference on Learning Representations (ICLR)*, 2017.

[7] Foret, P., et al. "Sharpness-Aware Minimization for Efficiently Improving Generalization." *International Conference on Learning Representations (ICLR)*, 2021.

**Limitations:**

yes

**Strengths And Weaknesses:**

*Strengths*
- Comparing methods at identical task-loss levels elegantly disentangles learning speed from trajectory geometry. This matched-loss evaluation is a rigorous and non-standard practice in the catastrophic forgetting literature.
- The four dynamical archetypes provide an intuitive, practical classification for optimization behaviors. Furthermore, the explicit distinction between "suppression" and "steering" cleanly explains why gradient noise merely halts learning while SVD actively reorients updates.
- The documented failure in the full-parameter stress test is highly valuable and honestly reported. It transparently delineates the exact operating boundaries where geometric regularization becomes ineffective.

*Weaknesses*
- The core insight prioritizing update direction over magnitude lacks novelty, as prior works have already explored low-dimensional subspaces [1] and direction-dependent regularization [2]. Furthermore, recent literature has already operationalized steering LoRA away from dominant pre-trained directions [3, 4, 5].
- The paper insufficiently acknowledges its deep connection to the flatness/sharpness literature [6, 7], treating these foundational concepts only in passing. Maximizing singular-value entropy to favor "soft" directions is functionally identical to Sharpness-Aware Minimization's objective applied to forgetting.
- The irreversibility claim is overstated, as the limited testing with SGD and momentum may merely reflect an optimizer artifact rather than a fundamental landscape property. A larger learning rate could potentially escape these basins, challenging the claim of permanent trajectory lock-in.

---

> ### Author Rebuttal · Authors · 2026-03-28
>
> We thank the reviewer for the rigorous feedback. Our contribution is not the generic observation that update direction matters, but a matched-loss framework for separating suppression, partial widening, and sustained steering under narrow-task PEFT.
>
> **Q1 / Weakness 3: Optimizer boundary**
> Under AdamW, the same lock-in / reorientation contrast remains visible. As shown in **Rev wUD8, Table D**, at matched task loss the uncontrolled run remains geometrically compressed (Last10_Rank = 6.9), while directional control preserves a much wider spectrum (83.3) and much lower broad loss (3.602 vs. 5.489).
>
> Relatedly, **Rev E4sz, Table A** directly tests the “larger LR may escape” concern: broad loss worsens from 3.042 to 3.682 as LR increases, while Last10_Rank rises only from 3.3 to 7.1, still far below the SVD case (99.0). We therefore do not claim strict irreversibility, but practical lock-in in the evaluated regime: larger step size changes speed and severity without producing a clean geometric escape.
>
> **Q2 / Weaknesses 1–2: Relation to prior work and recent steering methods**
> We agree that the paper should position itself more explicitly relative to [1][2][6][7] and recent steering methods [3][4][5]. Our claim is not that SVD Control is the best safe-learning method; rather, SVD serves here as a controlled directional probe under matched-loss comparison.
>
> During rebuttal, we ran a [4]-style MiLoRA proxy, since this was the only accessible implementation in time. Its matched-loss result is shown below. Here Broad@ℓ* denotes the broad loss evaluated at a shared matched task-loss anchor ℓ* by linear interpolation.
>
> Setting | Broad@ℓ* | Rank@ℓ* | Signature
> :--|:--:|:--:|:--
> Random Init | 3.246 | 4.0 | Geometric Deadlock
> MiLoRA proxy | 3.218 | 70.7 | Init Bias / Partial Widening
> SVD Control | 2.944 | 103.2 | Spectral Bifurcation
>
> **Table H.** MiLoRA proxy under matched-loss evaluation (N=256, ℓ* = 0.706).
>
> As shown in **Table H**, a [4]-style geometry-biased initialization is intermediate between random initialization and sustained steering: it widens the effective spectrum, but does not reach the same degree of bifurcation as explicit SVD control. Our distinction from [6][7] is therefore not simply that softer directions help, but that matched-loss probes separate braking, initialization bias / partial widening, and trajectory-level reorientation. In this sense, [1] shows that fine-tuning can operate in a low-dimensional subspace; our contribution is to ask which orientations within that subspace are harmful for broad retention under narrow-task PEFT.
>
> **Q3: Breakdown across capacity and data regime**
> To identify where the breakdown begins, we first ablated nominal LoRA capacity under fixed alpha/r = 2 and evaluated it at matched task loss:
>
> | r | Method | Broad@ℓ* | Rank@ℓ* | Final Broad Loss | Last10_Rank |
> | :---: | :--- | :---: | :---: | :---: | :---: |
> | 32  | Base | 3.145 | 2.7   | 3.438 | 2.9   |
> | 32  | SVD  | 2.933 | 29.4  | 2.929 | 29.4  |
> | 128 | Base | 3.183 | 3.7   | 3.493 | 3.8   |
> | 128 | SVD  | 2.966 | 77.5  | 2.939 | 98.9  |
> | 256 | Base | 3.211 | 3.8   | 3.753 | 4.5   |
> | 256 | SVD  | 2.937 | 135.9 | 2.914 | 166.5 |
>
> **Table G.** LoRA rank sweep at matched task loss, ℓ* = 0.770.
>
> Table G shows no gradual transition: effective rank stays below 5 across $r = 32$–$256$, versus $136.8$ for full fine-tuning. This reveals the bottleneck is structural (the product parameterization), not a matter of trainable subspace size. In $\Delta W = BA$,
>
> $$\frac{\partial \mathcal{L}}{\partial B} = \frac{\partial \mathcal{L}}{\partial \Delta W} A^\top,\quad \frac{\partial \mathcal{L}}{\partial A} = B^\top \frac{\partial \mathcal{L}}{\partial \Delta W},$$
>
> so once $A,B$ are confined to low-rank subspaces, gradient components orthogonal to them are zeroed out. Updates stay within $\mathrm{col}(B)\otimes \mathrm{row}(A)$. Larger $r$ provides nominal capacity, but if dynamics trap the model in a low-rank attractor, the extra dimensions remain inaccessible to gradient updates.
>
> For regime boundaries, **Rev wUD8, Table C** shows that Full-FT remains far more distributed than LoRA across tested data scales, while **Rev yZ9i, Table E** shows that scarcity changes the severity of broad harm but not the existence of the compressed LoRA regime itself.
>
> **Q4: Practical value of the probes**
> The probes are primarily diagnostic, but they already support three immediate uses:
> (1) geometric monitoring, where rank collapse reveals when training remains trapped in a compressed regime;
> (2) low-overhead directional control, where **Rev yZ9i, Table F** shows that steering only a small subset of layers remains affordable;
> (3) auxiliary online warning, where **Rev E4sz, Table B** shows that the smoothed cumulative Abs-Harm signal provides a usable early-stop or rescue-trigger window before final broad degradation unfolds.

---

> > ### Author Rebuttal · Reviewer_837X · 2026-04-03
> >
> > Thank you for the detailed rebuttal. The additional experiments and clarifications addressed my main concerns, especially around optimizer dependence, the learning-rate question, and the practical role of the probes, so I am comfortable moving my score to **4 (weak accept)**. My only remaining reservation is that the novelty claim is still not fully clean to me: the paper’s central message seems close to existing ideas that update direction matters more than magnitude and that flatter/softer directions help generalization, so I would still encourage the authors to state more explicitly what the paper adds beyond that prior literature and to sharpen the contribution in the final version.

---

> > > ### Author Response · Authors · 2026-04-04
> > >
> > > Thank you again for the careful follow-up and for the helpful suggestion on novelty positioning.
> > >
> > > We agree that the final version should sharpen more explicitly what is new here. Our intended claim is not that update direction matters in general, nor that flatter or softer directions help generalization per se. Rather, we provide a matched-loss geometric diagnostic framework that makes it possible to distinguish suppression from genuine trajectory-level reorientation during narrow-task PEFT. In this framing, SVD Control is used as a controlled diagnostic probe rather than presented as an optimal safe-learning method.
> > >
> > > We will revise the paper accordingly by tempering the novelty language, clarifying the relationship to flatness/sharpness analyses and prior steering methods, and sharpening the contribution around trajectory-level diagnosis under matched-loss evaluation.

---

### Official Review · Reviewer_yZ9i · 2026-03-11

**Soundness:** 3
**Presentation:** 3
**Significance:** 3
**Originality:** 4
**Overall Recommendation:** 4
**Confidence:** 1

**Summary:**

Under narrow-task fine-tuning, magnitude-based stability indicators such as parameter drift often fail to predict the degradation of broad capabilities. A geometric perspective on fine-tuning dynamics is introduced, suggesting that capability loss may arise from trajectory-level phenomena in parameter space rather than from displacement magnitude alone. To analyze these dynamics, the authors introduce three objective-agnostic geometric probes—update drift, stable update rank, and directional coherence—which together define a magnitude–direction phase space for diagnosing different fine-tuning regimes. Building on these observations, the authors provide a mechanistic interpretation based on spectral bias in the curvature of the loss landscape. They hypothesize that anisotropic Hessian structure near initialization channels optimization updates toward a small set of stiff curvature directions, forming a low-rank attractor that leads to trajectory lock-in. Escaping this regime may require a spectral bifurcation, i.e., a qualitative reorientation of the update subspace toward softer curvature modes. Experimentally, the authors evaluate this hypothesis across multiple model scales, modalities, and narrow-task settings using LoRA-based parameter-efficient fine-tuning.

**Compliance With Llm Reviewing Policy:**

Affirmed.

**Key Questions For Authors:**

1. Following the failure case reported in Appendix G, is there an empirical scaling law or theoretical threshold relating dataset size and the required spectral regularization strength? More concretely, how can practitioners determine whether a fine-tuning dataset is sufficiently large to prevent the proposed “Structured Navigation” dynamics from being overridden by rapid memorization?
2.  Exact SVD operations have cubic time complexity and may introduce nontrivial computational overhead for large-scale fine-tuning. Could you clarify whether applying SVD Control to billion-parameter LLMs requires specific algorithmic adjustments?
3. While the proposed dynamical archetypes provide strong theoretical intuition, the empirical validation currently relies on relatively simple baselines. Could comparisons with more recent PEFT methods designed to mitigate catastrophic forgetting or preserve structural properties be included, either in the current version or in a future revision?

**Limitations:**

yes

**Strengths And Weaknesses:**

Strengths
1. A geometric interpretation of capability degradation during fine-tuning is presented. By formalizing the Drift Paradox and introducing the concept of trajectory lock-in, a mechanistic perspective is provided in which optimization dynamics may concentrate updates along a restricted set of curvature directions in parameter space.
2. Fine-tuning dynamics are categorized into four distinct archetypes—Geometric Deadlock, Stochastic Stagnation, Inertial Braking, and Structured Navigation. This taxonomy systematically differentiates optimization behaviors across methods and offers a structured framework for analyzing and comparing fine-tuning regularization strategies.

Weaknesses
1. Although the complete failure of SVD Control under full-parameter fine-tuning with an extreme micro-dataset is transparently reported, the explanation that rapid memorization overrides geometric regularization remains largely empirical.
2. SVD Control requires computing singular values of the update matrices to minimize the negative entropy of the normalized spectrum. Since exact SVD operations carry nontrivial computational cost, applying this regularization at each optimization step may introduce efficiency bottlenecks.
3. The taxonomy table contrasts SVD Control with standard LoRA, Gradient Noise, and Null-Space Projection, which serve as useful baselines for illustrating the four archetypes. However, demonstrating the practical advantages of SVD Control for “Safe Learning” would benefit from comparisons with parameter-efficient fine-tuning methods explicitly designed to mitigate catastrophic forgetting or maintain orthogonality.

---

> ### Author Rebuttal · Authors · 2026-03-28
>
> We sincerely thank the reviewer for recognizing the geometric perspective, the Drift Paradox, and the dynamical taxonomy.
>
> **Q1 & Weakness 1: Is there an empirical scaling law or threshold relating dataset size and required spectral regularization?**
> We do not claim a closed-form scaling law or a single-variable threshold. In our view, the regime likely depends on multiple factors, especially data scale, narrow-task homogeneity, and task–pretraining similarity. Our new sweep clarifies the practical trend across representative data regimes.
>
> | Setting | Final Broad Loss | Last10_Rank | Note |
> | :--- | :---: | :---: | :--- |
> | n64_base | 4.647 | 3.6 | Extreme scarcity |
> | n64_svd_s10 | 2.906 | 87.3 | Wide spectrum retained |
> | n256_base | 3.545 | 3.8 | Matched regime |
> | n256_svd_s10 | 2.948 | 99.0 | Wide spectrum retained |
> | n1024_base | 3.257 | 3.6 | Larger narrow set |
> | n1024_svd_s10 | 3.003 | 103.6 | Wide spectrum retained |
>
> *Table E. Data-size sweep across representative narrow-task regimes.*
>
> As shown in **Table E**, extreme scarcity clearly amplifies broad degradation. At the same time, increasing $N$ alone does not remove geometric collapse: across the tested range, the uncontrolled baseline remains highly compressed (Last10_Rank ≈ 3.6–3.8), whereas the paired SVD-steered runs retain both lower broad harm and a much wider effective spectrum. This suggests that stronger regularization is most likely needed in regimes that are simultaneously smaller-scale and more homogeneous, though we do not yet claim a quantitative threshold law. Practically, we would therefore not rely on dataset size alone; a more useful criterion is whether the baseline still shows simultaneous broad-loss growth and severe rank collapse, which indicates that structured navigation is still being overridden.
>
> **Q2 & Weakness 2: Do exact SVD operations introduce efficiency bottlenecks at 1B scale, and what algorithmic adjustments make them practical?**
> We agree that global exact SVD is computationally expensive, but global intervention is not required to obtain the geometric effect.
>
> To directly address efficiency, we profiled the **pure step time** on a single NVIDIA GPU, averaged over 200 optimization steps after warmup, while varying the number of intervened layers.
>
> | Setting | Pure Step Time (ms) | Time Overhead |
> | :--- | :---: | :---: |
> | Baseline | 285.10 | 0.00% |
> | SVD ($L=2$) | 291.23 | +2.15% |
> | SVD ($L=4$) | 299.98 | +5.22% |
> | SVD ($L=8$) | 329.26 | +15.49% |
> | SVD ($L=16$) | 364.07 | +27.70% |
> | SVD ($L=32$) | 444.91 | +56.06% |
> | SVD ($L=64$) | 619.81 | +117.40% |
>
> *Table F. Exact SVD pure step-time overhead across different numbers of intervened layers.*
>
> As shown in **Table F**, at our current 1B scale, steering a small randomly sampled subset of layers (e.g., $L=4$) remains affordable, adding only **+5.22%** overhead. The corresponding practical adjustment is therefore not global exact control, but small-subset steering in the low-rank adapter space; full-layer exact SVD is best viewed as an analytical stress test rather than the default deployment path. The absolute timings differ from those in the manuscript because this profiling was rerun under a cleaner pure-step protocol in a different hardware environment, and we will clarify this measurement protocol in the revision.
>
> **Q3 & Weakness 3: Could comparisons with more recent PEFT methods be included?**
> We agree that these connections should be made more explicit. While we do not claim a complete head-to-head benchmark across recent PEFT methods in the rebuttal, we now have two scope-extending comparisons beyond the original baseline set.
>
> First, in **Rev 837X, Table H**, a MiLoRA-inspired proxy reaches Broad@ℓ* 3.218 and Rank@ℓ* 70.7, intermediate between Random Init (3.246, 4.0) and explicit SVD steering (2.944, 103.2). This suggests that geometry-biased initialization can partially widen the effective update spectrum, but does not achieve the same degree of bifurcation as sustained steering.
>
> Second, in **Rev E4sz, Table I**, we extended the analysis beyond vanilla LoRA to selective parameter tuning. There, unconstrained selective tuning still collapses into a rigid low-rank degradation path, while structured SVD control substantially widens the effective update spectrum and improves broad retention at matched task loss. This is especially relevant here because it shows that the collapse-versus-reorientation distinction is not confined to low-rank architectural constraints alone.
>
> Taken together, these additions do not yet constitute a full benchmark of recent PEFT methods, but they do extend the framework in two directions already: toward geometry-biased initialization and toward non-LoRA selective parameter tuning. We will revise the paper to position methods such as MiLoRA and related PEFT approaches more explicitly within this geometry-based view.

---

> > ### Author Rebuttal · Reviewer_yZ9i · 2026-04-01
> >
> > Thank you for the detailed response. This addresses my concerns, and I will keep my score to recommend accept.

---

### Official Review · Reviewer_wUD8 · 2026-03-12

**Soundness:** 3
**Presentation:** 3
**Significance:** 3
**Originality:** 3
**Overall Recommendation:** 4
**Confidence:** 3

**Summary:**

This paper investigates the geometric mechanisms underlying the degradation of general capabilities during narrow-task fine-tuning of Large Language Models (LLMs). The authors identify a phenomenon termed "Trajectory Lock-in," where optimization trajectories, regardless of the optimizer or hyperparameters, collapse onto a shared low-dimensional curve characterized by a trade-off between task loss reduction and general capability loss.

The study challenges the conventional reliance on parameter drift magnitude as a stability proxy, introducing the "Drift Paradox": identical Euclidean displacement can lead to divergent generalization outcomes depending on the trajectory's orientation. To address this, the authors propose three objective-agnostic geometric probes (Drift, Stable Rank, and Directional Coherence) and a spectral regularization method based on SVD. This method induces a "Spectral Bifurcation," reorienting the update trajectory away from "stiff" curvature modes associated with degradation, thereby preserving general capabilities while maintaining task performance.

**Compliance With Llm Reviewing Policy:**

Affirmed.

**Final Justification:**

Based on the paper and the rebuttal, my final justification is **Weak Accept**.

**Key Questions For Authors:**

1. **Generalizability of Mechanisms**: The observed "Trajectory Lock-in" is primarily documented in Llama-family models under LoRA settings. To what extent do you anticipate this phenomenon persists in fundamentally different architectures, such as Mixture-of-Experts (MoE) models, or in full-parameter fine-tuning regimes outside of the extreme data-scarcity setting (N=64)?

2. **Cost-Benefit Analysis**: The SVD regularization introduces an approximate 6.6% time overhead per step. In large-scale production environments, how do you evaluate the trade-off between this computational cost and the gains in model safety/generalization? Are there strategies to reduce this overhead for deployment?

3. **Role of Data Homogeneity**: Beyond the spectral bias of the Hessian, to what extent does the homogeneity of the narrow training data contribute to the "Trajectory Lock-in"? Could diverse data distributions naturally mitigate this geometric confinement without explicit regularization?

4. **Validity of Hessian Proxies**: The study uses a generic Hessian (H_gen) as a proxy for the task-specific Hessian (H_task). How sensitive are the results to the degree of overlap between the eigenspaces of H_gen and H_task across different tasks? Does a low overlap significantly degrade the efficacy of the spectral steering?

**Limitations:**

Yes.

**Strengths And Weaknesses:**

## Soundness

• **Strengths**: The empirical validation is robust, covering multiple model scales (1B to 8B) and modalities (reasoning, coding, storytelling) within the Llama family. The experimental design effectively contrasts the proposed method against strong baselines, including Null-space projection and gradient noise injection, clearly distinguishing between "stability by suppression" and "stability by steering."

• **Weaknesses**: The mechanistic explanation relies heavily on the Hessian matrix at initialization. While supported by experiments, the paper lacks a deeper discussion on how the Hessian evolves dynamically during fine-tuning and whether this evolution impacts the validity of the initial spectral bias hypothesis. Additionally, the method's failure in full-parameter fine-tuning on extremely small datasets (N=64) highlights specific boundary conditions that warrant further theoretical exploration.

## Presentation

• **Strengths**: The paper is exceptionally well-structured. Figures 1 and 2 intuitively illustrate the four dynamical archetypes (Geometric Deadlock, Stochastic Stagnation, Inertial Braking, Structured Navigation), making complex geometric concepts accessible. The phase-space visualization effectively communicates the core contributions.

• **Weaknesses**: Certain mathematical definitions, particularly the stochastic estimation of "Stable Rank," are primarily detailed in the appendix. Integrating more intuitive explanations of these core probes into the main text would improve readability for a broader audience.

## Significance

• **Strengths**: The work addresses a critical practical challenge: mitigating catastrophic forgetting while preserving plasticity. By reframing stability from a scalar magnitude problem to a geometric orientation problem, it offers a novel perspective that could influence future safe fine-tuning protocols. The concept of the "Drift Paradox" fundamentally questions current heuristics in the field.

• **Weaknesses**: While the SVD regularizer is effective, the authors candidly acknowledge it serves more as a diagnostic probe than an optimized production algorithm. Its generalizability to very large models (70B+) or architectures beyond LoRA (e.g., full fine-tuning or MoE) remains an open question.

## Originality

• **Strengths**: The introduction of "Trajectory Lock-in" and "Spectral Bifurcation" provides fresh terminology and conceptual frameworks for understanding fine-tuning dynamics. Unlike prior landscape analysis that focuses on convergence points, this work uniquely analyzes the geometry of the optimization trajectory itself.

---

> ### Author Rebuttal · Authors · 2026-03-28
>
> We sincerely thank the reviewer for the careful and constructive evaluation, and especially for recognizing the cross-scale and cross-modal empirical validation, the distinction between stability by suppression and stability by steering, and the originality of the trajectory lock-in / spectral bifurcation perspective.
>
> **Q1: Generalizability of mechanisms**
> We address this in three boundaries: architecture, optimizer, and scope beyond vanilla LoRA.
>
> **(a) Architecture boundary: LoRA vs. Full-FT across data scales**
>
> | Regime | Setting | Final Broad Loss | Last10_Rank | Note |
> | :--- | :--- | :---: | :---: | :--- |
> | N=64 | LoRA base | 4.647 | 3.6 | Severe lock-in |
> | N=64 | Full-FT base | 2.946 | 137.0 | Distributed regime |
> | N=256 | LoRA base | 3.545 | 3.8 | Matched regime |
> | N=256 | Full-FT base | 2.973 | 136.8 | Distributed regime |
> | N=4096 | LoRA base | 3.289 | 4.2 | Persistent lock-in |
> | N=4096 | Full-FT base | 2.926 | 118.9 | Distributed regime |
>
> *Table C. Architectural boundary across tested data scales.*
>
> As shown in **Table C**, Full-FT remains in a much more distributed-update regime than LoRA in our tested settings and does not exhibit the same severe spectrum collapse. We do not interpret this as universal safety of Full-FT. Our narrower claim is that, in the evaluated regimes, the strongest lock-in is structurally tied to the low-rank PEFT bottleneck. Beyond vanilla LoRA, a scope-extending selective-tuning check in Rev E4sz, **Table I** shows the same collapse-vs.-reorientation pattern after removing the architectural rank constraint. For MoE architectures, we do not claim direct evidence here, and we will state this explicitly as a limitation in the revision.
>
> **(b) Optimizer boundary: AdamW**
> Here Broad Loss (@ℓ*) denotes the broad loss evaluated at a shared matched task-loss anchor ℓ* by linear interpolation.
>
> | Setting | Broad Loss (@ℓ*) | Final Task Loss | Last10_Rank | Note |
> | :--- | :---: | :---: | :---: | :--- |
> | AdamW Base | 5.489 | 0.237 | 6.9 | Lock-in |
> | AdamW + SVD | 3.602 | 0.257 | 83.3 | Reorientation |
>
> *Table D. AdamW matched-loss comparison (N=256), ℓ* = 0.262.*
>
> As shown in **Table D**, the phenomenon remains visible under AdamW. The uncontrolled run is still geometrically compressed, while directional control preserves a much wider effective spectrum and substantially lower broad loss at matched task loss. This argues against a purely SGD-specific explanation in our tested regime.
>
> **Q2: Cost-benefit analysis of SVD control**
> As reported in **paper Table 3**, SVD Control adds about 6.6% time per step with no additional peak VRAM. We therefore view exact full-layer SVD as an analytical stress test rather than the default deployment path. The deployment-relevant path is small-subset steering, which preserves the geometric effect without requiring global exact control.
>
> **Q3: Role of data homogeneity**
> As shown in Rev yZ9i, **Table E**, baseline LoRA remains geometrically collapsed across representative data sizes (Last10_Rank ≈ 3.6–3.8), while broad harm is substantially worse under extreme scarcity (Final Broad Loss 4.647 at N=64 vs. 3.257 at N=1024). We therefore interpret homogeneity/scarcity as an amplifier of lock-in severity, but not the sole cause: the geometric bottleneck persists even as dataset size increases.
>
> **Q4: Validity of Hessian proxies and eigenspace overlap**
> To probe the boundary of using $H_{gen}$ as a proxy for $H_{task}$, we computed an empirical Fisher / gradient-subspace proxy for a representative deep layer (`v_proj`, >$10^6$ parameters) in both domains. The normalized Top-$K$ overlap is small in absolute value but clearly non-random: Top-5 = 1.96%, Top-10 = 2.30%, and Top-20 = 2.79%. For reference, the expected overlap between two random 20-dimensional subspaces in a >$10^6$-dimensional space is only about 0.0019%, so the observed Top-20 overlap is roughly three orders of magnitude above random.
>
> This matches our empirical picture: the proxy and task subspaces are largely distinct, preserving task plasticity, yet they still share a small but measurable intersection that provides a usable steering signal. The natural boundary is therefore low-overlap regimes: as the overlap approaches zero, proxy-based steering should weaken. We will report these Top-$K$ measurements in the revision and state low-overlap regimes explicitly as a boundary of the approach.

---

> > ### Author Rebuttal · Reviewer_wUD8 · 2026-04-03
> >
> > Thank you for your response, I decided to keep my score.

---

### Official Review · Reviewer_E4sz · 2026-03-22

**Soundness:** 2
**Presentation:** 3
**Significance:** 2
**Originality:** 3
**Overall Recommendation:** 4
**Confidence:** 2

**Summary:**

This paper presents a geometric analysis of narrow downstream fine-tuning of large language models. It first identifies a trajectory lock-in phenomenon, where fine-tuning methods follow a similar degradation path characterized by decreasing task loss and increasing general loss. The paper then argues that model drift alone is insufficient as an indicator of stability, showing that similar magnitudes of parameter drift can lead to different general losses. Finally, it introduces two additional metrics, directional coherence and stable rank, to characterize the geometric dynamics of fine-tuning. Experimental verification demonstrates that structured spectral steering enables a bifurcation away from this shared degradation path.

**Compliance With Llm Reviewing Policy:**

Affirmed.

**Final Justification:**

Overall, the paper presents an interesting analytical perspective on downstream task fine-tuning, distinguishing between optimization dynamics that reduce magnitude and those that reorient directions. My original concerns were twofold. First, regarding the novelty and the availability of concrete next steps derived from the analysis, the rebuttal better summarizes and clarifies the contributions, allowing me to better recognize them. It also provides an example of using Abs-Harm as a warning signal during training. Second, regarding the extension beyond LoRA, the rebuttal includes a preliminary experiment on selective tuning by updating the MLP matrices in the final layers. I appreciate the authors’ efforts in conducting this additional study, and the provided evidence appears convincing. Therefore, I will raise my score from 3 to 4 and lean toward acceptance of the paper.

**Key Questions For Authors:**

Besides the weakness shown in the above section, please also see the following questions:

Q1: What downstream datasets are used in the experiments? Does the domain similarity between the pretrained model and the downstream tasks affect the results? For example, how does the result change when the pretrained model is trained on general text, while the downstream task involves specialized domains such as healthcare reports?

Q2: For Type III inertial braking cases, would simply increasing the learning rate be effective, given that the directions are highly coherent?

Q3: Given the analysis presented in this paper, what concrete next steps can the community take to improve LLM adaptation performance?

**Limitations:**

This paper focuses solely on LoRA. It remains unclear whether the findings generalize to full fine-tuning or to substantially different parameter-efficient fine-tuning methods, such as prompt tuning. In addition, the experiments focus only on LLaMA and do not consider other types of LLMs, such as mixture-of-experts architectures like Mixtral.

**Strengths And Weaknesses:**

Strength:

- The paper presents an interesting and novel perspective on the trade-off between downstream task performance and generalization.

- Overall, the paper is well structured and easy to follow, especially given that it is an analysis-focused work, and the motivations and observations are clearly presented.

Weakness:

- The trajectory lock-in phenomenon, where decreasing task loss is often accompanied by increasing general loss, appears to be a well-known observation that has been shown in prior work.

- In the right column of Lines 433–435, the paper concludes that these findings suggest future methods should prioritize directional control over scalar constraints. However, it does not provide preliminary insights, designs, or results on how this could be realized, particularly in comparison to SVD-based regularization.

-  Given the abundant parameter-efficient fine-tuning (PEFT) methods, the analysis in this paper is only conducted on LoRA, It remains unsure if the observations can be generalized to other types of PEFT methods, such as prompt tuning, or selective parameter tuning [1].

[1] 2025 CVPR Lessons and Insights from a Unifying Study of Parameter-Efficient Fine-Tuning (PEFT) in Visual Recognition

---

> ### Author Rebuttal · Authors · 2026-03-28
>
> We thank the reviewer for recognizing the novelty of our perspective and the analysis-focused framing.
> Our contribution is a geometric diagnostic framework for narrow fine-tuning: at comparable task loss, different fine-tuning variants can still produce very different broad-loss outcomes because they follow different update geometries.
>
> **Q1: Downstream datasets and domain similarity**
> Our main narrow-task experiments use GSM8K as the core setting, with MBPP and TinyStories as cross-task validation on code and story generation. We did not perform a controlled domain-similarity sweep, so we do not make a strong causal claim about domain gap alone. What we can say is that the same degradation pattern appears across math, code, and story, suggesting that it is not confined to one narrow-task distribution.
>
> We also rewrote GSM8K into a unified instruction-style JSON format to reduce simple surface-form overlap. We view this as contamination control rather than a controlled domain-gap variable.
>
> **Q2: Would simply increasing the learning rate escape the basin?**
> We compare methods at a shared task-loss anchor ℓ*, and compute Broad Loss (@ℓ*) by linear interpolation at the first crossing.
>
> | Setting | Broad Loss (@ℓ*) | Final Task Loss | Last10_Rank | Note |
> | :--- | :---: | :---: | :---: | :--- |
> | Base (LR=5e-3) | 3.042 | 0.750 | 3.3 | Reference lock-in |
> | Base (LR=1e-2) | 3.250 | 0.614 | 3.8 | Paper default |
> | Base (LR=2e-2) | 3.477 | 0.419 | 4.8 | Larger step |
> | Base (LR=5e-2) | 3.682 | 0.280 | 7.1 | Very large step |
> | SVD probe | 2.944 | 0.765 | 99.0 | Reorientation |
>
> *Table A. LR sweep at matched task loss (SGD, N=256), ℓ* = 0.706.*
>
> As shown in **Table A**, larger LR does not improve broad retention. Broad loss worsens steadily as LR increases, while effective rank remains highly compressed. Although Last10_Rank rises from 3.3 to 7.1, it remains far below the SVD probe (99.0), which is more consistent with faster movement within the same stiff subspace than with a clean departure. Larger step sizes mainly accelerate motion along the same coherent directions rather than inducing geometric escape.
>
> **Q3 / Weakness 2: Concrete next steps for the community**
> Two immediate directions follow from our results.
>
> First, online directional probes can be used as training-time warning signals. In particular, a smoothed cumulative Abs-Harm can serve as an early-warning, early-stop, or rescue-trigger signal.
>
> | Seed (Baseline) | Stopping Step | Train Loss at Stop | Broad Loss at Stop | Final Broad Loss |
> | :--- | :---: | :---: | :---: | :---: |
> | Seed 42 | 380 | 0.414 | 4.004 | 4.929 |
> | Seed 123 | 320 | 0.492 | 3.823 | 4.321 |
> | Seed 2024 | 400 | 0.397 | 3.677 | 4.368 |
>
> *Table B. Early-warning heuristic from smoothed cumulative Abs-Harm.*
>
> As shown in **Table B**, this signal provides a usable warning window before final broad degradation.
>
> Second, future adaptation methods should prioritize directional control, not only scalar magnitude control. Our results distinguish stability by suppression from stability by steering: shrinking or diffusing updates may preserve broad capability by slowing learning, while structured control can improve retention without sacrificing task progress.
>
> **Weakness 1: Novelty**
> We agree that the task-general trade-off itself is not new. Our contribution is more specific: we identify a shared degradation path under narrow fine-tuning, formulate the Drift Paradox showing that similar drift can still lead to very different broad outcomes, introduce three objective-agnostic geometric probes to measure this structure, and use an SVD-based diagnostic probe to show that the key is not merely reducing update magnitude, but reorienting the update subspace.
>
> **Weakness 3: Generalization beyond LoRA**
> We thank the reviewer for pointing to selective parameter tuning. To test whether our geometric diagnostic framework extends beyond vanilla LoRA, we conducted a scope-extending check using selective tuning: we removed the architectural rank constraint and selectively unfroze full-rank MLP matrices in the final layers, then evaluated the baseline and SVD probe under the same matched-loss protocol, using the lowest shared anchor ℓ* = 0.740.
>
> | Setting (Selective Tuning) | Broad Loss (@ℓ*=0.740) | Final Broad Loss | Stable Rank $w_{stable}(\Delta W_t)$ |
> | :--- | :---: | :---: | :---: |
> | Baseline | 3.836 | 3.994 | 5.7 |
> | SVD regularization | **3.574** | **3.606** | **275.6** |
>
> *Table I. Scope-extending check on selective tuning.*
>
> As shown in **Table I**, the same qualitative pattern persists beyond LoRA. Unconstrained selective tuning still collapses into a rigid low-rank degradation path, while the structured SVD probe widens the effective update spectrum and improves broad retention at matched task performance. We present this as evidence that the framework transfers beyond vanilla LoRA, while not claiming universality to all PEFT families or MoE architectures such as Mixtral.

---

> > ### Author Rebuttal · Reviewer_E4sz · 2026-04-03
> >
> > Thank you to the authors for the response. The rebuttal addresses my main concerns, particularly regarding the novelty, the concrete next steps, and the extension beyond LoRA. Therefore, I will raise my score from 3 to 4.

---

### Decision · Program_Chairs · 2026-04-30

**Decision:**

Accept (regular)

**Comment:**

This paper studies why narrow task adaptation can harm broad capabilities in large language models and argues that parameter drift alone is an unreliable indicator of this degradation. The authors analyze adaptation through the geometry of optimization trajectories and identify a recurring trajectory lock in pattern in which different methods follow a similar curve of lower task loss and higher general loss. To characterize this behavior, the paper introduces three geometric probes, namely update drift, stable rank, and directional coherence, along with an online Abs-Harm signal. The paper further proposes a spectral view in which anisotropic curvature near initialization channels updates into stiff directions, and suggests that escaping this behavior requires a qualitative reorientation of the update subspace. Across several models and multiple narrow task settings, the experiments show that an SVD based regularizer can move training away from the shared degradation path while preserving task performance better than baseline controls.

The reviews are positive about the clarity of the presentation, the strength of the empirical study, and the value of the geometric framing, especially the drift paradox, the dynamical archetypes, and the matched loss evaluation protocol. Reviewers also appreciate the reporting of failure cases and the distinction between suppression of learning and steering of updates. The main concerns are that some aspects of the core intuition are related to prior work on directionality, subspaces, and sharpness, that the mechanistic argument relies on curvature at initialization, and that generalization beyond LoRA and to larger or different adaptation settings remains insufficiently established. There are also questions about computational overhead, comparisons to other baselines, and whether some claims around lock in may depend on optimizer choices. The rebuttal helps clarify the scope of the claims, and the paper makes a meaningful contribution by synthesizing these ideas into a coherent framework with supporting evidence. I am inclined to recommend acceptance because the paper offers a valuable perspective, introduces practical diagnostic tools, and provides empirical robustness and novelty.